# Projection-Free Methods for Stochastic Simple Bilevel Optimization with Convex Lower-level Problem

**Jincheng Cao**
ECE Department
UT Austin
jinchengcao@utexas.edu

**Ruichen Jiang**
ECE Department
UT Austin
rjiang@utexas.edu

**Nazanin Abolfazli**
SIE Department
The University of Arizona
nazaninabolfazli@arizona.edu

**Erfan Yazdandoost Hamedani**
SIE Department
The University of Arizona
erfany@arizona.edu

**Aryan Mokhtari**
ECE Department
UT Austin
mokhtari@austin.utexas.edu

## Abstract

In this paper, we study a class of stochastic bilevel optimization problems, also known as stochastic simple bilevel optimization, where we minimize a smooth stochastic objective function over the optimal solution set of another stochastic convex optimization problem. We introduce novel stochastic bilevel optimization methods that locally approximate the solution set of the lower-level problem via a stochastic cutting plane, and then run a conditional gradient update with variance reduction techniques to control the error induced by using stochastic gradients. For the case that the upper-level function is convex, our method requires $\tilde{\mathcal{O}}(\max\{1/\epsilon_f^2, 1/\epsilon_g^2\})$ stochastic oracle queries to obtain a solution that is $\epsilon_f$-optimal for the upper-level and $\epsilon_g$-optimal for the lower-level. This guarantee improves the previous best-known complexity of $\mathcal{O}(\max\{1/\epsilon_f^4, 1/\epsilon_g^4\})$. Moreover, for the case that the upper-level function is non-convex, our method requires at most $\tilde{\mathcal{O}}(\max\{1/\epsilon_f^3, 1/\epsilon_g^3\})$ stochastic oracle queries to find an $(\epsilon_f, \epsilon_g)$-stationary point. In the finite-sum setting, we show that the number of stochastic oracle calls required by our method are $\tilde{\mathcal{O}}(\sqrt{n}/\epsilon)$ and $\tilde{\mathcal{O}}(\sqrt{n}/\epsilon^2)$ for the convex and non-convex settings, respectively, where $\epsilon = \min\{\epsilon_f, \epsilon_g\}$.

## 1 Introduction

An important class of bilevel optimization problems is simple bilevel optimization in which we aim to minimize an upper-level objective function over the solution set of a lower-level problem [1–4]. Recently this class of problems has attracted great attention in machine learning society due to their applications in continual learning [5], hyper-parameter optimization [6, 7], meta-learning [8, 9], and reinforcement learning [10, 11]. Motivated by large-scale learning problems, in this paper, we are particularly interested in the stochastic variant of the simple bilevel optimization where the upper and lower-level objective functions are the expectations of some random functions with unknown distributions and are accessible only through their samples. Hence, the computation of the objective function values or their gradients is not computationally tractable. Specifically, we focus on the *stochastic simple bilevel problem* defined as

$$\min_{\mathbf{x} \in \mathbb{R}^d} \quad f(\mathbf{x}) = \mathbb{E}[\tilde{f}(\mathbf{x}, \theta)] \qquad \text{s.t.} \quad \mathbf{x} \in \arg\min_{\mathbf{z} \in \mathcal{Z}} g(\mathbf{z}) = \mathbb{E}[\tilde{g}(\mathbf{z}, \xi)], \tag{1}$$

37th Conference on Neural Information Processing Systems (NeurIPS 2023).

Table 1: Results on stochastic simple bilevel optimization. The abbreviations "SC", "C", and "NC" stand for "strongly convex", "convex", and "non-convex", respectively. Note that $\epsilon = \min\{\epsilon_f, \epsilon_g\}$

| References | Type | Upper level | Lower level | | Convergence | | Sample Complexity |
|---|---|---|---|---|---|---|---|
| | | Objective $f$ | Objective $g$ | Feasible set $\mathcal{Z}$ | Upper level | Lower level | |
| aR-IP-SeG [16] | Stochastic | C, Lipschitz | C, Lipschitz | Closed | $\mathcal{O}(\max\{1/\epsilon_f^4, 1/\epsilon_g^4\})$ | | $\mathcal{O}(1/\epsilon^4)$ |
| **Algorithm1** | Stochastic | C, smooth | C, smooth | Compact | $\tilde{\mathcal{O}}(\max\{1/\epsilon_f^2, 1/\epsilon_g^2\})$ | | $\tilde{\mathcal{O}}(1/\epsilon^2)$ |
| **Algorithm1** | Stochastic | NC, smooth | C, smooth | Compact | $\tilde{\mathcal{O}}(\max\{1/\epsilon_f^3, 1/\epsilon_g^3\})$ | | $\tilde{\mathcal{O}}(1/\epsilon^3)$ |
| **Algorithm2** | Finite-sum | C, smooth | C, smooth | Compact | $\tilde{\mathcal{O}}(\max\{1/\epsilon_f, 1/\epsilon_g\})$ | | $\tilde{\mathcal{O}}(\sqrt{n}/\epsilon)$ |
| **Algorithm2** | Finite-sum | NC, smooth | C, smooth | Compact | $\tilde{\mathcal{O}}(\max\{1/\epsilon_f^2, 1/\epsilon_g^2\})$ | | $\tilde{\mathcal{O}}(\sqrt{n}/\epsilon^2)$ |

where $\mathcal{Z}$ is compact and convex and $\tilde{f}, \tilde{g} : \mathbb{R}^d \to \mathbb{R}$ are continuously differentiable functions on an open set containing $\mathcal{Z}$, and $\theta$ and $\xi$ are some independent random variables drawn from some possibly unknown probability distributions. As a result, the functions $f, g : \mathbb{R}^d \to \mathbb{R}$ are also continuously differentiable functions on $\mathcal{Z}$. We assume that $g$ is convex but not necessarily strongly convex, and hence the solution set of the lower-level problem in (1) is in general not a singleton. We also study the finite sum version of the above problem where both functions can be written as the average of $n$ component functions, i.e., $f(\mathbf{x}) = (1/n) \sum_{i=1}^n \tilde{f}(\mathbf{x}, \theta_i)$ and $g(\mathbf{z}) = (1/n) \sum_{i=1}^n \tilde{g}(\mathbf{z}, \xi_i)$.

The main challenge in solving problem (1), which is inherited from its deterministic variant, is the absence of access to the feasible set, i.e., the lower-level solution set. This issue eliminates the possibility of using any projection-based or projection-free methods. There have been some efforts to overcome this issue in the deterministic setting (where access to $f$ and $g$ and their gradients is possible), including [12–15], however, there is little done on the stochastic setting described above. In fact, the only work that addresses the stochastic problem in (1) is [16], where the authors present an iterative regularization-based stochastic extra gradient algorithm and show that it requires $\mathcal{O}(1/\epsilon_f^4)$ and $\mathcal{O}(1/\epsilon_g^4)$ queries to the stochastic gradient of the upper-level and lower-level function, respectively, to obtain a solution that is $\epsilon_f$-optimal for the upper-level and $\epsilon_g$-optimal for the lower-level. We improve these bounds and also extend our results to nonconvex settings.

**Contributions.** In this paper, we present novel projection-free stochastic bilevel optimization methods with tight non-asymptotic guarantees for both upper and lower-level problems. At each iteration, the algorithms use a small number of samples to build unbiased and low variance estimates and construct a cutting plane to locally approximate the solution set of the lower-level problem and then combine it with a Frank-Wolfe-type update on the upper-level objective. Our methods require careful construction of the cutting plane so that with high probability it contains the solution set of the lower-level problem, which is obtained by selecting proper function and gradient estimators to achieve the obtained optimal convergence guarantees. Next, we summarize our main theoretical results for the proposed Stochastic Bilevel Conditional Gradient methods for Infinite and Finite sample settings denoted by SBCGI and SBCGF, respectively.

- (Stochastic setting) We show that SBCGI (Algorithm 1), in the convex setting, finds a solution $\hat{\mathbf{x}}$ that satisfies $f(\hat{\mathbf{x}}) - f^* \le \epsilon_f$ and $g(\hat{\mathbf{x}}) - g^* \le \epsilon_g$ with probability $1 - \delta$ within $\mathcal{O}(\log(d/\delta\epsilon)/\epsilon^2)$ stochastic oracle queries, where $\epsilon = \min\{\epsilon_f, \epsilon_g\}$, $f^*$ is the optimal value of problem (1) and $g^*$ is the optimal value of the lower-level problem. Moreover, in the non-convex setting, it finds $\hat{\mathbf{x}}$ satisfying $\mathcal{G}(\hat{\mathbf{x}}) \le \epsilon_f$ and $g(\hat{\mathbf{x}}) - g^* \le \epsilon_g$ with probability $1 - \delta$ within $\mathcal{O}((\log(d/\delta\epsilon))^{3/2}/\epsilon^3)$ stochastic oracle queries, where $\mathcal{G}(\hat{\mathbf{x}})$ is the Frank-Wolfe (FW) gap.

- (Finite-sum setting) We show that SBCGF (Algorithm 2), in the convex setting, finds $\hat{\mathbf{x}}$ that satisfies $f(\hat{\mathbf{x}}) - f^* \le \epsilon_f$ and $g(\hat{\mathbf{x}}) - g^* \le \epsilon_g$ with probability $1 - \delta$ within $\mathcal{O}(\sqrt{n}(\log(1/\delta\epsilon))^{3/2}/\epsilon)$ stochastic oracle queries, where $n$ is the number of samples of finite-sum problem. Moreover, in the nonconvex setting, it finds $\hat{\mathbf{x}}$ that satisfies $\mathcal{G}(\hat{\mathbf{x}}) \le \epsilon_f$ and $g(\hat{\mathbf{x}}) - g^* \le \epsilon_g$ with probability $1 - \delta$ within $\mathcal{O}(\sqrt{n}\log(1/\delta\epsilon)/\epsilon^2)$ stochastic oracle queries.

## 1.1 Related work

**General stochastic bilevel.** In a general format of stochastic bilevel problems, the upper-level function $f$ also depends on an extra variable $y \in \mathbb{R}^p$ which also affects the lower-level objective,

$$\min_{\mathbf{x} \in \mathbb{R}^d, \mathbf{y} \in \mathbb{R}^p} f(\mathbf{x}, \mathbf{y}) = \mathbb{E}[\tilde{f}(\mathbf{x}, \mathbf{y}, \theta)] \qquad \text{s.t. } \mathbf{x} \in \arg\min_{\mathbf{z} \in \mathcal{Z}} g(\mathbf{z}, \mathbf{y}) = \mathbb{E}[\tilde{g}(\mathbf{z}, \mathbf{y}, \xi)]. \tag{2}$$

There have been several works including [10, 17–21] on solving the general stochastic bilevel problem (2). However, they only focus on the setting where the lower-level problem is strongly convex, i.e., $g(\mathbf{z}, \mathbf{y})$ is strongly convex with respect to $\mathbf{z}$ for any value of $\mathbf{y}$. In fact, (2) with a convex lower-level problem is known to be NP-hard [22]. Hence, the results of these works are not directly comparable with our work as we focus on a simpler setting, but our assumption on the lower-level objective function is weaker and it only requires the function to be convex.

**Deterministic simple bilevel.** There have been some recent results on non-asymptotic guarantees for the deterministic variant of problem (1). The BiG-SAM algorithm was presented in [14], and it was shown that its lower-level objective error converges to zero at a rate of $\mathcal{O}(1/t)$, while the upper-level error asymptotically converges to zero. In [15], the authors achieved the first non-asymptotic rate for both upper- and lower-level problems by introducing an iterative regularization-based method which achieves an $(\epsilon_f, \epsilon_g)$-optimal solution after $\mathcal{O}(\max\{1/\epsilon_f^4, 1/\epsilon_g^4\})$ iterations. In [12], the authors proposed a projection-free method for deterministic simple bilevel problems that has a complexity of $\mathcal{O}(\max\{1/\epsilon_f, 1/\epsilon_g\})$ for convex upper-level and complexity of $\mathcal{O}(\max\{1/\epsilon_f^2, 1/(\epsilon_f \epsilon_g)\})$ for non-convex upper-level. Moreover, in [23] the authors presented a switching gradient method to solve simple bilevel problems with convex smooth functions for both upper- and lower-level problems with complexity $\mathcal{O}(1/\epsilon)$. However, all the above results are limited to the deterministic setting.

**General bilevel without lower-level strong convexity.** Recently, there are several recent works on general bilevel optimization problems without lower-level strong convexity including [24–27, 23]. However, they either have a weaker theoretical results like asymptotic convergence rate in [24] or have some additional assumptions. Specifically, in [25], the authors reformulated the problem as a constrained optimization problem and further assumes such problem to be a convex program. Moreover, in [26, 27], the authors in both papers assumed that the lower-level objective satisfies the PL inequality, while we assumed that the lower-level objective is convex. In [23], the authors used a looser convergence criterion that only guarantees convergence to a Goldstein stationary point. Since these works consider a more general class of problems, we argue that their theoretical results when applied to our setting are necessarily weaker.

## 2 Preliminaries

### 2.1 Motivating examples

*Example 1: Over-parameterized regression.* A general form of problem (1) is when the lower-level problem represents training loss and the upper-level represents test loss. The goal is to minimize the test loss by selecting one of the optimal solutions for the training loss [13]. An instance of that is the constrained regression problem, where we intend to find an optimal parameter vector $\boldsymbol{\beta} \in \mathbb{R}^d$ that minimizes the loss $\ell_{\mathrm{tr}}(\boldsymbol{\beta})$ over the training dataset $\mathcal{D}_{\mathrm{tr}}$. To represent some prior knowledge, we usually constrain $\boldsymbol{\beta}$ to be in some subsets $\mathcal{Z} \subseteq \mathbb{R}^d$, e.g., $\mathcal{Z} = \{\boldsymbol{\beta} \mid \|\boldsymbol{\beta}\|_1 \leq \lambda\}$ for some $\lambda > 0$ to induce sparsity. To handle multiple global minima, we adopt the over-parameterized approach, where the number of samples is less than the parameters. Although achieving one of these global minima is possible, not all optimal solutions perform equally on other datasets. Hence, we introduce an upper-level objective: the loss on a validation set $\mathcal{D}_{\mathrm{val}}$. This helps select a training loss optimizer that performs well on both training and validation sets. It leads to the following bilevel problem:

$$\min_{\boldsymbol{\beta} \in \mathbb{R}^d} f(\boldsymbol{\beta}) \triangleq \ell_{\mathrm{val}}(\boldsymbol{\beta}) \qquad \text{s.t.} \quad \boldsymbol{\beta} \in \arg\min_{\mathbf{z} \in \mathcal{Z}} g(\mathbf{z}) \triangleq \ell_{\mathrm{tr}}(\mathbf{z}) \tag{3}$$

In this case, both the upper- and lower-level losses are smooth and convex if $\ell$ is smooth and convex.

*Example 2: Dictionary learning.* Problem (1) also appears in lifelong learning, where the learner takes a series of tasks sequentially and tries to accumulate knowledge from past tasks to improve performance in new tasks. Here we focus on continual dictionary learning. The aim of dictionary

learning is to obtain a compact representation of the input data. Let $\mathbf{A} = \{\mathbf{a}_1, \ldots, \mathbf{a}_n\} \in \mathbb{R}^{m \times n}$ denote a dataset of $n$ points. We seek a dictionary $\mathbf{D} = [\mathbf{d}_1, \ldots, \mathbf{d}_p] \in \mathbb{R}^{m \times p}$ such that all data points $\mathbf{a}_i$ can be represented by a linear combination of basis vectors in $\mathbf{D}$ which can be cast as [28–31]:

$$\min_{\mathbf{D} \in \mathbb{R}^{m \times p}} \min_{\mathbf{X} \in \mathbb{R}^{p \times n}} \frac{1}{2n} \sum_{i \in \mathcal{N}} \|\mathbf{a}_i - \mathbf{D}\mathbf{x}_i\|_2^2 \qquad \text{s.t.} \ \ \|\mathbf{d}_j\|_2 \leq 1, j = 1, \ldots, p; \|\mathbf{x}_i\|_1 \leq \delta, i \in \mathcal{N}. \quad (4)$$

Moreover, we denote $\mathbf{X} = [\mathbf{x}_1, \ldots, \mathbf{x}_n] \in \mathbb{R}^{p \times n}$ as the coefficient matrix. In practice, data points usually arrive sequentially and the representation evolves gradually. Hence, the dictionary must be updated sequentially as well. Assume that we already have learned a dictionary $\hat{\mathbf{D}} \in \mathbb{R}^{m \times p}$ and the corresponding coefficient matrix $\hat{\mathbf{X}} \in \mathbb{R}^{p \times n}$ for the dataset $\mathbf{A}$. As a new dataset $\mathbf{A}' = \{\mathbf{a}'_1, \ldots, \mathbf{a}'_{n'}\}$ arrives, we intend to enrich our dictionary by learning $\tilde{\mathbf{D}} \in \mathbb{R}^{m \times q} (q > p)$ and the coefficient matrix $\tilde{\mathbf{X}} \in \mathbb{R}^{q \times n'}$ for the new dataset while maintaining good performance of $\tilde{\mathbf{D}}$ on the old dataset $\mathbf{A}$ as well as the learned coefficient matrix $\hat{\mathbf{X}}$. This leads to the following stochastic bilevel problem:

$$\min_{\tilde{\mathbf{D}} \in \mathbb{R}^{m \times q}} \min_{\tilde{\mathbf{X}} \in \mathbb{R}^{q \times n'}} f(\tilde{\mathbf{D}}, \tilde{\mathbf{X}}) \qquad \text{s.t.} \ \ \|\tilde{\mathbf{x}}_k\|_1 \leq \delta, k = 1, \ldots, n'; \ \ \tilde{\mathbf{D}} \in \underset{\|\tilde{\mathbf{d}}_j\|_2 \leq 1}{\arg\min} g(\tilde{\mathbf{D}}), \quad (5)$$

where $f(\tilde{\mathbf{D}}, \tilde{\mathbf{X}}) \triangleq \frac{1}{2n'} \sum_{k=1}^{n'} \|\mathbf{a}'_k - \tilde{\mathbf{D}}\tilde{\mathbf{x}}_k\|_2^2$ represents the average reconstruction error on the new dataset $\mathbf{A}'$, and $g(\tilde{\mathbf{D}}) \triangleq \frac{1}{2n} \sum_{i=1}^{n} \|\mathbf{a}_i - \tilde{\mathbf{D}}\hat{\mathbf{x}}_i\|_2^2$ represents the error on the old dataset $\mathbf{A}$. Note that we denote $\hat{\mathbf{x}}_i$ as the prolonged vector in $\mathbb{R}^q$ by appending zeros at the end. In problem (5), the upper-level objective is non-convex, while the lower-level loss is convex with multiple minima.

## 2.2 Assumptions and definitions

Next, we formally state the assumptions required in this work.

**Assumption 2.1.** *$\mathcal{Z}$ is convex and compact with diameter $D$, i.e., $\forall \mathbf{x}, \mathbf{y} \in \mathcal{Z}$, we have $\|\mathbf{x} - \mathbf{y}\| \leq D$.*

**Assumption 2.2.** *The upper-level stochastic function $\tilde{f}$ satisfies the following conditions:*

*(i) $\nabla \tilde{f}$ is Lipschitz with constant $L_f$, i.e., $\forall \mathbf{x}, \mathbf{y} \in \mathcal{Z}, \forall \theta, \|\nabla \tilde{f}(\mathbf{x}, \theta) - \nabla \tilde{f}(\mathbf{y}, \theta)\| \leq L_f \|\mathbf{x} - \mathbf{y}\|$.*

*(ii) The stochastic gradients noise is sub-Gaussian, $\mathbb{E}[\exp\{\|\nabla \tilde{f}(\mathbf{x}, \theta) - \nabla f(\mathbf{x})\|^2 / \sigma_f^2\}] \leq \exp\{1\}$.*

**Assumption 2.3.** *The lower-level stochastic function $\tilde{g}$ satisfies the following conditions:*

*(i) $g$ is convex and $\nabla \tilde{g}$ is $L_g$-Lipschitz, i.e., $\forall \mathbf{x}, \mathbf{y} \in \mathcal{Z}, \forall \xi, \|\nabla \tilde{g}(\mathbf{x}, \xi) - \nabla \tilde{g}(\mathbf{y}, \xi)\| \leq L_g \|\mathbf{x} - \mathbf{y}\|$.*

*(ii) The stochastic gradients noise is sub-Gaussian, $\mathbb{E}[\exp\{\|\nabla \tilde{g}(\mathbf{x}, \xi) - \nabla g(\mathbf{x})\|^2 / \sigma_g^2\}] \leq \exp\{1\}$.*

*(iii) The stochastic functions noise is sub-Gaussian, $\mathbb{E}[\exp\{|\tilde{g}(\mathbf{x}, \xi) - g(\mathbf{x})|^2 / \sigma_l^2\}] \leq \exp\{1\}$.*

*Remark* 2.1. Assumptions 2.1 and 2.3(i) imply that $\tilde{g}$ is Lipschitz continuous on an open set containing $\mathcal{Z}$ with some constant $L_l$, i.e., for all $\mathbf{x}, \mathbf{y} \in \mathcal{Z}$, we have $|\tilde{g}(\mathbf{x}) - \tilde{g}(\mathbf{y})| \leq L_l \|\mathbf{x} - \mathbf{y}\|$.

In the paper, we denote $g^* \triangleq \min_{\mathbf{z} \in \mathcal{Z}} g(\mathbf{z})$ and $\mathcal{X}_g^* \triangleq \arg\min_{\mathbf{z} \in \mathcal{Z}} g(\mathbf{z})$ as the optimal value and the optimal solution set of the lower-level problem, respectively. Note that by Assumption 2.3, the set $\mathcal{X}_g^*$ is nonempty, convex, and compact, but typically not a singleton as $g$ potentially has multiple minima on $\mathcal{Z}$. Furthermore, we denote $f^*$ and $\mathbf{x}^*$ as the optimal value and an optimal solution of problem (1), which are assured to exist since $f$ is continuous and $\mathcal{X}_g^*$ is compact.

**Definition 2.1.** *When $f$ is convex, a point $\hat{\mathbf{x}} \in \mathcal{Z}$ is $(\epsilon_f, \epsilon_g)$-optimal if $f(\hat{\mathbf{x}}) - f^* \leq \epsilon_f$ and $g(\hat{\mathbf{x}}) - g^* \leq \epsilon_g$. When $f$ is non-convex, $\hat{\mathbf{x}} \in \mathcal{Z}$ is $(\epsilon_f, \epsilon_g)$-optimal if $\mathcal{G}(\hat{\mathbf{x}}) \leq \epsilon_f$ and $g(\hat{\mathbf{x}}) - g^* \leq \epsilon_g$, where $\mathcal{G}(\hat{\mathbf{x}})$ is the FW gap [32, 33] defined as $\mathcal{G}(\hat{\mathbf{x}}) \triangleq \max_{\mathbf{s} \in \mathcal{X}_g^*} \{\langle \nabla f(\hat{\mathbf{x}}), \hat{\mathbf{x}} - \mathbf{s} \rangle\}$.*

## 3 Algorithms

**Conditional gradient for simple bilevel optimization.** A variant of the conditional gradient (CG) method for solving bilevel problems has been introduced in [12] which uses a cutting plane idea [34]

to approximate the solution set of the lower-level problem denoted by $\mathcal{X}_g^*$. More precisely, if one has access to $\mathcal{X}_g^*$, it is possible to run the FW update with stepsize $\gamma_t$ as

$$\mathbf{x}_{t+1} = (1 - \gamma_t)\mathbf{x}_t + \gamma_t \mathbf{s}_t, \qquad \text{where} \quad \mathbf{s}_t = \arg\min_{\mathbf{s} \in \mathcal{X}_g^*} \langle \nabla f(\mathbf{x}_t), s \rangle \tag{6}$$

However, the set $\mathcal{X}_g^*$ is not explicitly given and the above method is not implementable. In [12], the authors suggested the use of the following set: $\mathcal{X}_t = \{\mathbf{s} \in \mathcal{Z} : \langle \nabla g(\mathbf{x}_t), \mathbf{s} - \mathbf{x}_t \rangle \leq g(\mathbf{x}_0) - g(\mathbf{x}_t)\}$ instead of the set $\mathcal{X}_g^*$ in the FW update given in (6). Note that $\mathbf{x}_0$ is selected in a way that $g(\mathbf{x}_0) - g^*$ is smaller than $\epsilon_g/2$, and such a point can be efficiently computed. A crucial property of the above set is that it always contains the solution set of the lower-level problem denoted by $\mathcal{X}_g^*$. This can be easily verified by the fact that for any $\mathbf{x}_g^*$ in $\mathcal{X}_g^*$ we have $\mathbf{x}_g^* \in \mathcal{Z}$ and

$$\langle \nabla g(\mathbf{x}_t), \mathbf{x}_g^* - \mathbf{x}_t \rangle \leq g(\mathbf{x}_g^*) - g(\mathbf{x}_t) \leq g(\mathbf{x}_0) - g(\mathbf{x}_t), \tag{7}$$

where the second inequality holds as $g(\mathbf{x}_0) \geq g(\mathbf{x}_g^*)$. As shown in [12], this condition is sufficient to show that if one follows the update in (6) with $\mathcal{X}_t$ instead of $\mathcal{X}_g^*$, the iterates will converge to the optimal solution. However, this framework is not applicable to the stochastic setting as we cannot access the functions or their gradients. Next, we present our main idea to address this delicate issue.

**Random set for the subproblem.** A natural idea to address stochasticity is to replace all gradients and functions with their stochastic estimators for both the subproblem in (6), i.e., $\nabla f(\mathbf{x}_t)$, as well as the construction of the cutting plane $\mathcal{X}_t$, i.e., $g(\mathbf{x}_t)$, and $\nabla g(\mathbf{x}_t)$. However, this simple idea fails since the set $\mathcal{X}_t$ may no longer contain the solution set $\mathcal{X}_g^*$. More precisely, if $\hat{g}_t$ and $\widehat{\nabla g}_t$ are unbiased estimators of $g(\mathbf{x}_t)$ and $\nabla g(\mathbf{x}_t)$, respectively, for the following approximation set

$$\mathcal{X}_t' = \{\mathbf{s} \in \mathcal{Z} : \langle \widehat{\nabla g}_t, \mathbf{s} - \mathbf{x}_t \rangle \leq g(\mathbf{x}_0) - \hat{g}_t\} \tag{8}$$

we can not argue that it contains $\mathcal{X}_g^*$, as the second inequality in (7) does not hold, i.e., $\langle \widehat{\nabla g}(\mathbf{x}_t), \mathbf{x}_g^* - \mathbf{x}_t \rangle \leq \hat{g}(\mathbf{x}_g^*) - \hat{g}(\mathbf{x}_t) \nleq g(\mathbf{x}_0) - \hat{g}(\mathbf{x}_t)$. In the appendix, we numerically illustrate this point.

To address this issue, we tune the cutting plane by only moving it but not rotating it, i.e., adding another term to tolerate the noise from stochastic estimates. We introduce the stochastic cutting plane

$$\hat{\mathcal{X}}_t = \{\mathbf{s} \in \mathcal{Z} : \langle \widehat{\nabla g}_t, \mathbf{s} - \mathbf{x}_t \rangle \leq g(\mathbf{x}_0) - \hat{g}_t + K_t\}, \tag{9}$$

where $\widehat{\nabla g}_t$ and $\hat{g}_t$ are gradient and function value estimators, respectively, that we formally define later. In the above expression, the addition of the term $K_t$, which is a sequence of constants converging to zero as $t \to \infty$, allows us to ensure that with high probability the random set $\hat{\mathcal{X}}_t$ contains all optimal solutions of the lower-level problem. Choosing suitable values for the sequence $K_t$ is a crucial task. If we select a large value for $K_t$ then the probability of $\hat{\mathcal{X}}_t$ containing $\mathcal{X}_g^*$ goes up at the price allowing points with larger values $g$ in the set. As a result, once we perform an update similar to the one in (6), the lower-level function value could increase significantly. On the other hand, selecting small values for $K_t$ would allow us to show that the lower level objective function is not growing, while the probability of $\hat{\mathcal{X}}_t$ containing $\mathcal{X}_g^*$ becomes smaller which could even lead to a case that the set becomes empty and the bilevel problem becomes infeasible.

*Remark* 3.1. How to compute $g(\mathbf{x}_0)$? In the finite sum setting, we can accurately compute $g(\mathbf{x}_0)$, and the additional cost of $n$ function evaluations will be dominated by the overall complexity. In the stochastic setting, we could use a large batch of samples to compute $g(\mathbf{x}_0)$ with high precision at the beginning of the process. This additional operation will not affect the overall sample complexity of the proposed method, as the additional cost is negligible compared to the overall sample complexity. Specifically, we need to take a batch size of $b = \tilde{\mathcal{O}}(\epsilon^{-2})$ to estimate $\hat{g}(\mathbf{x}_0)$. Using the Hoeffding inequality for subgaussian random variables, we have the following bound: $|\hat{g}(\mathbf{x}_0) - g(\mathbf{x}_0)| \leq \sqrt{2}\sigma_l(T+1)^{-\omega/2}\sqrt{\log(2/\delta)}$, with a probability of at least $1 - \delta$, where $T$ is the maximum number of iterations. Comparing this with Lemma 4.1.3, we can further derive: $|\hat{g}(\mathbf{x}_0) - g(\mathbf{x}_0)| \leq \sqrt{2}\sigma_l(T+1)^{-\omega/2}\sqrt{\log(2/\delta)} \leq \sqrt{2}(2L_lD + \frac{3^\omega}{3^\omega-1}\sigma_l)(t+1)^{-\omega/2}\sqrt{\log(6/\delta)}$, with a probability of at least $1 - \delta$ for all $0 \leq t \leq T$. Consequently, the introduced error term would be absorbed in $K_{0,t}$ and will not affect any parts of the analysis.

**Variance reduced estimators.** As mentioned above, a key point in the design of our stochastic bilevel algorithms is to select $K_t$ properly such that $\hat{\mathcal{X}}_t$ contains $\mathcal{X}_g^*$ with high probability, for all $t \geq 0$. To

---

**Algorithm 1** SBCGI

---

1: **Input**: Target accuracy: $\epsilon_f, \epsilon_g > 0$, probability $\delta > 0$, step size: $\alpha_t, \beta_t, \rho_t, \gamma_t > 0$
2: **Initialization**: Initialize $\mathbf{x}_0 \in \mathcal{Z}$ such that $g(\mathbf{x}_0) - g^* \leq \epsilon_g/2$
3: **for** $t = 0, \ldots, T$ **do**
4:     **if** $t = 0$ **then**
5:         $\widehat{\nabla f}_t = \nabla \tilde{f}(\mathbf{x}_t, \theta_t), \widehat{\nabla g}_t = \nabla \tilde{g}(\mathbf{x}_t, \xi_t), \hat{g}_t = \tilde{g}(\mathbf{x}_t, \xi_t)$
6:     **else**
7:         Update the estimate of $\nabla f$, $\widehat{\nabla f}_t = (1-\alpha_t)\widehat{\nabla f}_{t-1} + \nabla \tilde{f}(\mathbf{x}_t, \theta_t) - (1-\alpha_t)\nabla \tilde{f}(\mathbf{x}_{t-1}, \theta_t)$
8:         Update the estimate of $\nabla g$, $\widehat{\nabla g}_t = (1-\beta_t)\widehat{\nabla g}_{t-1} + \nabla \tilde{g}(\mathbf{x}_t, \xi_t) - (1-\beta_t)\nabla \tilde{g}(\mathbf{x}_{t-1}, \xi_t)$
9:         Update the estimate of $g$, $\hat{g}_t = (1 - \rho_t)\hat{g}_{t-1} + \tilde{g}(\mathbf{x}_t, \xi_t) - (1 - \rho_t)\tilde{g}(\mathbf{x}_{t-1}, \xi_t)$
10:    **end if**
11:    Compute $\mathbf{s}_t \in \arg\min_{\mathbf{s} \in \mathcal{X}_t} \{\widehat{\nabla f}_t^\top \mathbf{s}\}$ where $\mathcal{X}_t = \{\mathbf{s} \in \mathcal{Z} : \langle \widehat{\nabla g}_t, \mathbf{s} - \mathbf{x}_t \rangle \leq g(\mathbf{x}_0) - \hat{g}_t + K_t\}$
12:    Update the variable $\mathbf{x}_{t+1} = (1 - \gamma_{t+1})\mathbf{x}_t + \gamma_{t+1}\mathbf{s}_t$
13: **end for**

---

achieve such a guarantee, we first need to characterize the error of our gradient and function value estimators. More precisely, suppose that for our function estimator we have that $P(|\hat{g}_t - g(\mathbf{x}_t)| \leq K_{0,t}) \geq 1 - \delta'$ and for the gradient estimator we have $P(\|\widehat{\nabla g}_t - \nabla g(\mathbf{x}_t)\| \leq K_{1,t}) \geq 1 - \delta'$, for some $\delta' \in (0, 1)$. Then, by setting $K_t = K_{0,t} + DK_{1,t}$, we can guarantee that the conditions required for the inequalities in (7) hold with probability at least $(1 - 2\delta')$.

Using simple sample average estimators would not allow for the selection of a diminishing $K_t$, as the variance is not vanishing, but by using proper variance-reduced estimators the variance of the estimators vanishes over time and eventually, we can send $K_t$ to zero. In this section, we focus on two different variance reduction estimators. For the stochastic setting in (1) we use the STOchastic Recursive Momentum estimator (STORM), proposed in [35], and for the finite-sum setting, we utilize the Stochastic Path-Integrated Differential EstimatoR (SPIDER) proposed in [36]. If $\mathbf{v}_{t-1}$ is the gradient estimator of STORM at time $t - 1$, the next estimator is computed as

$$\mathbf{v}_t = (1 - \alpha_t)\mathbf{v}_{t-1} + \nabla \tilde{f}(\mathbf{x}_t, \theta_t) - (1 - \alpha_t)\nabla \tilde{f}(\mathbf{x}_{t-1}, \theta_t), \tag{10}$$

where $\nabla \tilde{f}(\mathbf{x}, \theta)$ is the stochastic gradient evaluated at $\mathbf{x}$ with sample $\theta$. The main advantage of the above estimator is that it can be implemented even with one sample per iteration. Unlike STORM, for the SPIDER estimator, we need a larger batch of samples per update. More precisely, if we consider $\mathbf{v}_{t-1}$ as the estimator of SPIDER for $\nabla f(\mathbf{x}_t)$, it is updated according to

$$\mathbf{v}_t = \nabla f_{\mathcal{S}}(\mathbf{x}_t) - \nabla f_{\mathcal{S}}(\mathbf{x}_{t-1}) + \mathbf{v}_{t-1}, \tag{11}$$

where $\nabla f_{\mathcal{S}}(\mathbf{x}) = (1/S) \sum_{i \in \mathcal{S}} \nabla f(\mathbf{x}, \theta_i)$ is the average sub-sampled stochastic gradient computed using samples that are in the set $\mathcal{S}$. As we will discuss later, in the finite sum case that we use SPIDER, the size of batch $S$ depends on $n$ which is the number of component functions. We delay establishing a high probability error bound for these estimators to section 4.1.

### 3.1 Conditional gradient algorithms with random sets: stochastic and finite-sum

Next, we present our Stochastic Bilevel Conditional Gradient method for Infinite sample case abbreviated by SBCGI for solving (1) and its finite sum variant denoted by SBCGF. In both cases, we first find a point $\mathbf{x}_0$ that satisfies $g(\mathbf{x}_0) - g^* \leq \epsilon_g/2$, for some accuracy $\epsilon_g$. The cost of finding such a point is negligible compared to the cost of the main algorithm as we discuss later. At each iteration $t$, we first update the gradient estimator of the upper-level and the function and gradient estimators of the lower-level problem. In SBCGI, we follow the STORM idea as described in steps 4-6 of Algorithm 1, while in SBCGF, we use the SPIDER technique as presented in steps 7-10 of Algorithm 2.In the case of SBCGF, we need to compute the exact gradient and function values once every $q$ iteration as presented in steps 4-6 of Algorithm 2. Once the estimators are updated, we can define the random set $\hat{\mathcal{X}}_t$ as in (9) and solve the following subproblem over the set $\hat{\mathcal{X}}_t$,

$$\mathbf{s}_t = \arg\min_{\mathbf{s} \in \hat{\mathcal{X}}_t} \langle \widehat{\nabla f}_t, \mathbf{s} \rangle, \tag{12}$$

---

**Algorithm 2** SBCGF

---

1: **Input**: Target accuracy: $\epsilon_f, \epsilon_g > 0$, probability accuracy: $\delta_0, \delta_1 > 0$, step size: $\gamma_t > 0$
2: **Initialization**: Initialize $\mathbf{x}_0 \in \mathcal{Z}$ such that $g(\mathbf{x}_0) - g^* \leq \epsilon_g/2$
3: **for** $t = 0, \ldots, T$ **do**
4:     **if** $\mathrm{mod}(t, q) = 0$ **then**
5:         Set $\widehat{\nabla f}_t = \nabla f(\mathbf{x}_t), \widehat{\nabla g}_t = \nabla g(\mathbf{x}_t), \hat{g}_t = g(\mathbf{x}_t)$
6:     **else**
7:         Draw $S$ samples
8:         Update the estimate of $\nabla f$ as $\widehat{\nabla f}_t = \widehat{\nabla f}_{t-1} + \nabla f_\mathcal{S}(\mathbf{x}_t) - \nabla f_\mathcal{S}(\mathbf{x}_{t-1})$
9:         Update the estimate of $\nabla g$ as $\widehat{\nabla g}_t = \widehat{\nabla g}_{t-1} + \nabla g_\mathcal{S}(\mathbf{x}_t) - \nabla g_\mathcal{S}(\mathbf{x}_{t-1})$
10:       Update the estimate of $g$ as $\hat{g}_t = \hat{g}_{t-1} + g_\mathcal{S}(\mathbf{x}_t) - g_\mathcal{S}(\mathbf{x}_{t-1})$
11:     **end if**
12:     Compute $\mathbf{s}_t \in \arg\min_{\mathbf{s} \in \mathcal{X}_t} \{\widehat{\nabla f}_t^\top \mathbf{s}\}$ where $\mathcal{X}_t = \{\mathbf{s} \in \mathcal{Z} : \langle \widehat{\nabla g}_t, \mathbf{s} - \mathbf{x}_t \rangle \leq g(\mathbf{x}_0) - \hat{g}_t + K_t\}$
13:     Update the variable $\mathbf{x}_{t+1} = (1 - \gamma_{t+1})\mathbf{x}_t + \gamma_{t+1}\mathbf{s}_t$
14: **end for**

---

where $\widehat{\nabla f}_t$ is the unbiased estimator of $\nabla f(\mathbf{x}_t)$. Note that we implicitly assume that we have access to a linear optimization oracle that returns a solution of the subproblem in (12), which is standard for projection-free methods [32, 33]. In particular, if $\mathcal{Z}$ can be described by a system of linear inequalities, then problem (12) corresponds to a linear program and can be solved by a standard solver as we show in our experiments. Once, $\mathbf{s}_t$ is calculated we simply update the iterate

$$\mathbf{x}_{t+1} = (1 - \gamma_{t+1})\mathbf{x}_t + \gamma_{t+1}\mathbf{s}_t \tag{13}$$

with stepsize $\gamma_{t+1} \in [0, 1]$. The only missing part for the implementation of our methods is the choice of $K_t$ in the random set and the stepsize parameters. We address these points in the next section.

*Remark* 3.2. SBCGI can be implemented with a batch size as small as $S = 1$. However, this does not imply that the batch size "has to be" $S = 1$. In other words, the main advantage of SBCGI, compared to SBCGF, is its capability to be implemented with any mini-batch size, even as small as $S = 1$. Therefore, for SBCGI, the batch size can be set arbitrarily, whereas for SBCGF, it must be $\sqrt{n}$.

*Remark* 3.3. In the finite-sum setting, if the numbers of functions in the upper- and lower-level losses are different, we could simply modify SBCGF 2 by choosing $S_u = q_u = \sqrt{n_u}$ and $S_l = q_l = \sqrt{n_l}$, where $n_u$ and $n_l$ are the number of functions in the upper- and lower-level, respectively.

## 4 Convergence analysis

In this section, we characterize the sample complexity of our methods for stochastic and finite-sum settings. Before stating our results, we first characterize a high probability bound for the estimators of our algorithms, which are crucial in the selection of parameter $K_t$ and the overall sample complexity.

### 4.1 High probability bound for the error terms

To achieve a high probability bound, it is common to assume that the noise of gradient or function is uniformly bounded as in [36, 37], but such assumptions may not be realistic for most machine learning applications. Hence, in our analysis, we consider a milder assumption and assume the noise of function and gradient are sub-Gaussian as in Assumptions 2.2 and 2.3, respectively. Given these assumptions, we next establish a high probability error bound for the estimators in SBCGI.

**Lemma 4.1.** *Consider SBCGI in Algorithm 1 with parameters $\alpha_t = \beta_t = \rho_t = \gamma_t = 1/(t + 1)^\omega$ where $\omega \in (0, 1]$. If Assumptions 2.1, 2.2, and 2.3 are satisfied, for any $t \geq 1$ and $\delta \in (0, 1)$, with probability at least $1 - \delta$, for some absolute constant $c$, ($d$ is the number of dimension), we have*

$$\|\widehat{\nabla f}_t - \nabla f(\mathbf{x}_t)\| \leq c\sqrt{2}(2L_f D + \frac{3^\omega}{3^\omega - 1}\sigma_f)(t + 1)^{-\omega/2}\sqrt{\log(6d/\delta)}, \tag{14}$$

$$\|\widehat{\nabla g}_t - \nabla g(\mathbf{x}_t)\| \leq c\sqrt{2}(2L_g D + \frac{3^\omega}{3^\omega - 1}\sigma_g)(t + 1)^{-\omega/2}\sqrt{\log(6d/\delta)}, \tag{15}$$

$$|\hat{g}_t - g(\mathbf{x}_t)| \leq c\sqrt{2}(2L_l D + \frac{3^\omega}{3^\omega - 1}\sigma_l)(t + 1)^{-\omega/2}\sqrt{\log(6/\delta)}. \tag{16}$$

Lemma 4.1 shows that for any $\omega \in (0, 1]$, if we set $\alpha_t = \beta_t = \rho_t = \gamma_t = 1/(t+1)^\omega$, then with high probability the gradient and function approximation errors converge to zero at a sublinear rate of $\tilde{\mathcal{O}}(1/t^{\omega/2})$. Moreover, the above result characterizes the choice of $K_t$. Specifically, if we define $K_{1,t}$ as the upper bound in (15) and $K_{0,t}$ as the upper bound in (16), by setting $K_t = K_{0,t} + DK_{1,t}$, then with probability $(1 - \delta)$ the random set $\hat{\mathcal{X}}_t$ contains $\mathcal{X}_g^*$. Later, we show that $\omega = 1$ leads to the best complexity bound for the convex setting and $\omega = 2/3$ is the best choice for the nonconvex setting.

Next, we establish a similar result for the estimators in SBCGF.

**Lemma 4.2.** *Consider SBCGF with stepsize $\gamma$ and $S = q = \sqrt{n}$. If Assumptions 2.1-2.3 hold, for any $t \geq 1$ and $\delta \in (0, 1)$, with probability $1 - \delta$ we have $\|\widehat{\nabla f}_t - \nabla f(\mathbf{x}_t)\| \leq 4L_g D\gamma\sqrt{\log(12/\delta)}$, $\|\widehat{\nabla g}_t - \nabla g(\mathbf{x}_t)\| \leq 4L_g D\gamma\sqrt{\log(12/\delta)}$, and $|\hat{g}_t - g(\mathbf{x}_t)| \leq 4L_l D\gamma\sqrt{\log(12/\delta)}$.*

Similarly, for SBCGF, we set $K_{1,t} = 4L_g D\gamma\sqrt{\log(12/\delta)}$ and $K_{0,t} = 4L_l D\gamma\sqrt{\log(12/\delta)}$ and choose $K_t = K_{0,t} + DK_{1,t}$, then the random set $\hat{\mathcal{X}}_t$ contains $\mathcal{X}_g^*$ with probability $1 - \delta$.

Next, we formalize our claim about the random set with the above choice of $K_t$.

**Lemma 4.3.** *If $\mathcal{X}_g^*$ is the solution set of the lower-level problem and $\hat{\mathcal{X}}_t$ is the feasible set constructed by cutting plane at iteration t, then for any $t \geq 0$ and $\delta \in (0, 1)$, we have $\mathbf{P}(\mathcal{X}_g^* \subseteq \hat{\mathcal{X}}_t) \geq 1 - \delta$.*

This lemma shows all $\mathcal{X}_g^*$ is a subset of the constructed feasible set $\hat{\mathcal{X}}_t$ with a high probability of $1 - \delta$. Indeed, using a union bound one can show that the above statement holds for all iterations up to time $t$ with probability $1 - t\delta$.

## 4.2 Convergence and complexity results for the stochastic setting

Next, we characterize the iteration and sample complexity of the proposed method in SBCGI for the stochastic setting. First, we present the result for the case that $f$ is convex.

**Theorem 4.4** (Stochastic setting with convex upper-level). *Consider SBCGI in Algorithm 1 for solving problem* (1)*. Suppose Assumptions 2.1, 2.2, and 2.3 hold and $f$ is convex. If the stepsizes of SBCGI are selected as $\alpha_t = \beta_t = \rho_t = \gamma_t = (t+1)^{-1}$, and the cutting plane parameter is $K_t = c((2L_l D + \frac{3}{2}\sigma_l)\sqrt{2\log(6t/\delta)} + D(2L_g D + \frac{3}{2}\sigma_g)\sqrt{2\log(6td/\delta)})(t+1)^{-1/2}$, then after $T$ iterations,*

$$g(\mathbf{x}_T) - g^* \leq \frac{C_1\zeta}{\sqrt{T}} + \frac{L_g D^2 \log T}{T} + \frac{\epsilon_g}{2}, \quad f(\mathbf{x}_T) - f^* \leq \frac{C_2\zeta}{\sqrt{T}} + \frac{f(\mathbf{x}_0) - f^* + L_f D^2 \log T}{T}.$$

*with probability $1 - \delta$ for some absolute constants $C_1$ and $C_2$ and $\zeta := \sqrt{\log(6Td/\delta)}$.*

Theorem 4.4 shows a convergence rate of $\mathcal{O}(\sqrt{\log(Td/\delta)}/\sqrt{T})$. As a corollary, SBCGI returns an $(\epsilon_f, \epsilon_g)$-optimal solution with probability $1 - \delta$ after $\mathcal{O}(\log(d/\delta\epsilon)/\epsilon^2)$ iterations, where $\epsilon = \min\{\epsilon_f, \epsilon_g\}$. Since we use one sample per iteration, the overall sample complexity is also $\mathcal{O}(\log(d/\delta\epsilon)/\epsilon^2)$. Note that the iteration complexity and sample complexity of our method outperform the ones in [16], as they require $\mathcal{O}(1/\epsilon^4)$ iterations and sample to achieve the same guarantee.

*Remark* 4.1. The task of finding $\mathbf{x}_0$ which is equivalent to a single-level stochastic optimization problem requires $\mathcal{O}(1/\epsilon_g^2)$ iterations and samples. As a result, this additional cost does not affect the overall complexity of our method. The same argument also holds in the non-convex case.

**Theorem 4.5** (Stochastic setting with non-convex upper level). *Consider SBCGI for solving problem* (1)*. Suppose Assumptions 2.1-2.3 hold, $f$ is nonconvex, and define $\underline{f} = \min_{\mathbf{x} \in \mathcal{Z}} f(\mathbf{x})$. If the stepsizes of SBCGI are selected as $\alpha_t = \beta_t = \rho_t = (t+1)^{-2/3}$, $\gamma_t = (T+1)^{-2/3}$, and the cutting plane parameter is $K_t = c((2L_l D + \frac{3^{2/3}}{3^{2/3}-1}\sigma_l)\sqrt{2\log(6T/\delta)} + D(2L_g D + \frac{3^{2/3}}{3^{2/3}-1}\sigma_g)\sqrt{2\log(6Td/\delta)})(t+1)^{-1/3}$, then after $T$ iterations, there exists an iterate $\mathbf{x}_{t^*}$ in the set $\in \{\mathbf{x}_0, \mathbf{x}_1, \ldots, \mathbf{x}_{T-1}\}$ for which*

$$g(\mathbf{x}_{t^*}) - g^* \leq \frac{C_3\zeta + L_g D^2}{(T+1)^{1/3}} + \frac{\epsilon_g}{2}, \quad \mathcal{G}(\mathbf{x}_{t^*}) \leq \frac{f(\mathbf{x}_0) - \underline{f} + C_4\zeta + L_f D^2}{(T+1)^{1/3}}$$

*with probability $1 - \delta$ for some absolute constants $C_3$ and $C_4$ and $\zeta := \sqrt{\log(6Td/\delta)}$.*

As a corollary of Theorem 4.5, the number of iterations required to find an $(\epsilon_f, \epsilon_g)$-optimal solution can be upper bounded by $\mathcal{O}(\log(d/\delta\epsilon)^{3/2}/\epsilon^3)$, where $\epsilon = \min\{\epsilon_f, \epsilon_g\}$. We note that the dependence on the upper-level accuracy $\epsilon_f$ also matches that in the standard CG method for a single-level non-convex problem [33, 38]. Moreover, as we only need one stochastic oracle query per iteration, SBCGI only requires $\mathcal{O}(\log(d/\delta\epsilon)^{3/2}/\epsilon^3)$ stochastic oracle queries to find an $(\epsilon_f, \epsilon_g)$-optimal.

### 4.3 Convergence and complexity results for the finite-sum setting

Similarly, we present iteration and sample complexity for algorithm 2 under the finite-sum setting.

**Theorem 4.6** (Finite-sum setting with convex upper-level). *Consider SBCGF presented in Algorithm 2 for solving the finite-sum version of* (1). *Suppose Assumptions 2.1, 2.2, and 2.3 hold and $f$ is convex. If we set the stepsizes of SBCGF as $\gamma = \log T/T$, $S = q = \sqrt{n}$, and the cutting plane parameter as $K_t = 4D(L_l\sqrt{\log(12T/\delta)} + L_g D\sqrt{\log(12T/\delta)})\log T/T$, then after $T$ iterations,*

$$g(\mathbf{x}_T) - g^* \leq \frac{(C_5\zeta' + L_g D^2)\log T}{T} + \frac{\epsilon_g}{2}, \quad f(\mathbf{x}_T) - f^* \leq \frac{f(\mathbf{x}_0) - f^* + C_6\zeta'\log T}{T},$$

*with probability at least $1 - \delta$, for some absolute constant $C_5$ and $C_6$, and $\zeta' = \sqrt{\log(12T/\delta)}$.*

Theorem 4.6 implies the number of stochastic oracle queries is $\mathcal{O}(\log(1/\delta\epsilon)^{3/2}\sqrt{n}/\epsilon)$, where $\epsilon = \min\{\epsilon_f, \epsilon_g\}$, which matches the optimal sample complexity of single-level problems [39].

**Theorem 4.7** (Finite-sum setting with non-convex upper-level). *Consider SBCGF presented in Algorithm 2 for solving the finite-sum version of* (1). *Suppose Assumptions2.1-2.3 hold, and $f$ is non-convex. Define $\underline{f} = \min_{\mathbf{x} \in \mathcal{Z}} f(\mathbf{x})$. If the parameters of SBCGF are selected as $\gamma = 1/\sqrt{T}$, $S = q = \sqrt{n}$, and the cutting plane parameter is $K_t = 4D(L_l\sqrt{\log(12T/\delta)} + L_g D\sqrt{\log(12T/\delta)})/\sqrt{T}$, then after $T$ iterations, there exists an iterate $\mathbf{x}_{t^*} \in \{\mathbf{x}_0, \mathbf{x}_1, \ldots, \mathbf{x}_{T-1}\}$ for which,*

$$g(\mathbf{x}_{t^*}) - g^* \leq \frac{C_7\zeta' + L_g D^2}{T^{1/2}} + \frac{\epsilon_g}{2}, \quad \mathcal{G}(\mathbf{x}_{t^*}) \leq \frac{f(\mathbf{x}_0) - \underline{f} + C_8\zeta'}{T^{1/2}}$$

*with probability at least $1 - \delta$, for some absolute constants $C_7$ and $C_8$, and $\zeta' = \sqrt{\log(12T/\delta)}$.*

As a corollary of Theorem 4.7, the number of stochastic oracle queries is $\mathcal{O}(\log(1/\delta\epsilon)\sqrt{n}/\epsilon^2)$, where $\epsilon = \min\{\epsilon_f, \epsilon_g\}$, which matches the state-of-the-art single-level result $\mathcal{O}(\sqrt{n}/\epsilon^2)$ in [40]. SBCGF also improves the number of linear minimization oracle queries of SBCGI from $\mathcal{O}(1/\epsilon^2)$ to $\mathcal{O}(1/\epsilon)$ for convex upper-level and from $\mathcal{O}(1/\epsilon^3)$ to $\mathcal{O}(1/\epsilon^2)$ for non-convex upper-level.

## 5 Numerical experiments

In this section, we test our methods on two different stochastic bilevel optimization problems with real and synthetic datasets and compare them with other existing stochastic methods in [16] and [13].

**Over-parameterized regression.** We consider the bilevel problem corresponding to sparse linear regression introduced in (3). We apply the Wikipedia Math Essential dataset [30] which composes of a data matrix $\mathbf{A} \in \mathbb{R}^{n \times d}$ with $n = 1068$ samples and $d = 730$ features and an output vector $\mathbf{b} \in \mathbb{R}^n$. To ensure the problem is over-parameterized, we assign $1/3$ of the dataset as the training set $(\mathbf{A}_{\text{tr}}, \mathbf{b}_{\text{tr}})$, $1/3$ as the validation set $(\mathbf{A}_{\text{val}}, \mathbf{b}_{\text{val}})$ and the remaining $1/3$ as the test set $(\mathbf{A}_{\text{test}}, \mathbf{b}_{\text{test}})$. For both upper- and lower-level loss functions we use the least squared loss, and we set $\lambda = 10$. We compare the performance of our methods with the aR-IP-SeG method by [16] and the stochastic version of DBGD introduced by [13]. We employ CVX [41, 42] to solve the lower-level problem and the reformulation of the bilevel problem to obtain $g^*$ and $f^*$, respectively. We also include the additional cost of finding $\mathbf{x}_0$ in SBCGI and SBCGF in our comparisons.

In Figure 1(a)(b), we observe that SBCGF maintains a smaller lower-level gap than other methods and converges faster than the rest in terms of upper-level error. SBCGI has the second-best performance in terms of lower- and upper-level gaps, while aR-IP-SeG performs poorly in terms of both lower- and upper-level objectives. The performance of DBGD-sto for the upper-level objective is well, however, it underperforms in terms of lower-level error. In Figure 1(c), SBCGF, SBCGI, and DBGD-sto achieve almost equally small test errors, while aR-IP-SeG fails to achieve a low test error. Note

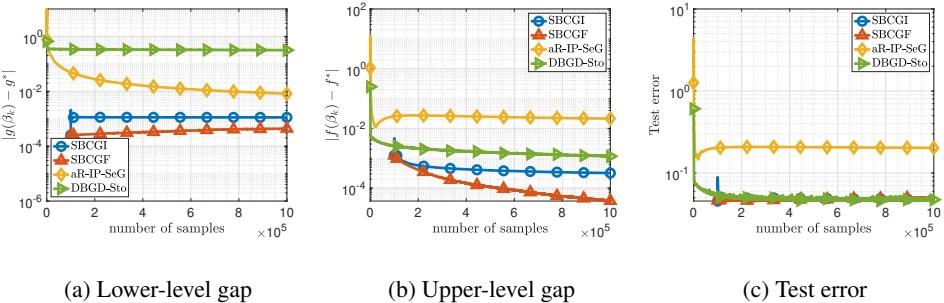

(a) Lower-level gap         (b) Upper-level gap         (c) Test error

Figure 1: Comparison of SBCGI, SBCGF, aR-IP-SeG, and DBGD-Sto for solving Problem (3)

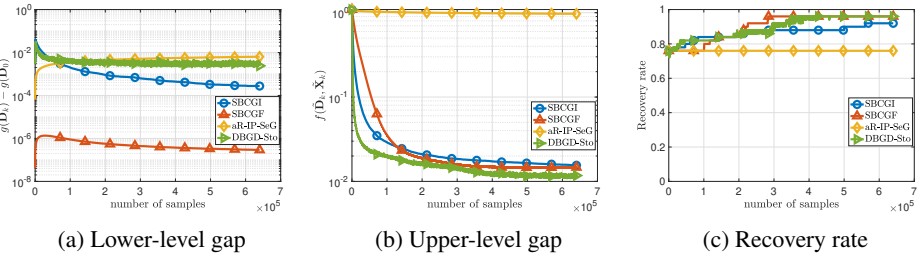

(a) Lower-level gap         (b) Upper-level gap         (c) Recovery rate

Figure 2: Comparison of SBCGI, SBCGF, aR-IP-SeG, and DBGD-Sto for solving Problem (5).

that after the initial stage, SBCGI increases slightly in terms of all the performance criteria, because SBCGI (1) only takes one sample per iteration and uses a decreasing step-size while SBCGF takes $\sqrt{n}$ samples per iteration and uses a small constant stepsize, demonstrating a more robust performance.

**Dictionary learning.** To test our methods on problems with non-convex upper-level we consider problem (5) on a synthetic dataset with a similar setup to [12]. We first construct the true dictionary $\tilde{\mathbf{D}}^* \in \mathbb{R}^{25\times 50}$ comprising of 50 basis vectors in $\mathbb{R}^{25}$. All entries of these basis vectors are drawn from the standard Gaussian distribution and then normalized to have unit $\ell_2$-norm. We also generate two more dictionaries $\mathbf{D}^*$ and $\mathbf{D}'^*$ consisting of 40 and 20 basis vectors in $\tilde{\mathbf{D}}^*$, respectively (thus they share at least 10 bases). These two datasets $\mathbf{A} = \{\mathbf{a}_1, \ldots, \mathbf{a}_{250}\}$ and $\mathbf{A}' = \{\mathbf{a}'_1, \ldots, \mathbf{a}'_{250}\}$ are constructed as $\mathbf{a}_i = \mathbf{D}^*\mathbf{x}_i + \mathbf{n}_i$, for $i = 1, \ldots, 250$, and $\mathbf{a}'_k = \mathbf{D}'^*\mathbf{x}'_k + \mathbf{n}'_k$, for $k = 1, \ldots, 250$, where $\{\mathbf{x}_i\}_{i=1}^{250}, \{\mathbf{x}'_k\}_{k=1}^{250}$ are coefficient vectors and $\{\mathbf{n}_i\}_{i=1}^{250}, \{\mathbf{n}'_k\}_{k=1}^{250}$ are random Gaussian noises. As neither $\mathbf{A}$ nor $\mathbf{A}'$ includes all the elements of $\tilde{\mathbf{D}}^*$, it is important to renew our dictionary by using the new dataset $\mathbf{A}'$ while maintaining the knowledge from the old dataset $\mathbf{A}$.

In our experiment, we initially solve the standard dictionary learning problem employing dataset $\mathbf{A}$, achieving the initial dictionary $\hat{\mathbf{D}}$ and coefficient vectors $\{\hat{\mathbf{x}}\}_{i=1}^{250}$. We define the lower-level objective as the reconstruction error on $\mathbf{A}$ using $\{\hat{\mathbf{x}}\}_{i=1}^{250}$, and the upper-level objective as the error on new dataset $\mathbf{A}'$. We compare our algorithms with aR-IP-SeG and DBGD (stochastic version), measuring performance with the recovery rate of true basis vectors. Note that a basis vector $\tilde{\mathbf{d}}_i^*$ in $\tilde{\mathbf{D}}^*$ is considered as successfully recovered if there exists $\tilde{\mathbf{d}}_j$ in $\tilde{\mathbf{D}}$ such that $|\langle \tilde{\mathbf{d}}_i^*, \tilde{\mathbf{d}}_j \rangle| > 0.9$ (for more details of the experiment setup see Appendix F). In Figure 2(a), we observe SBCGF converges faster than any other method regarding the lower-level objective. While SBCGI has the second-best performance in terms of the lower-level gap, aR-IP-SeG and DBGD-sto perform poorly compared with SBCGI and SBCGF. In Figures 2(b) and (c), we see that SBCGI, SBCGF, and DBGD-sto achieve good results in terms of the upper-level objective and the recovery rate. However, aR-IP-SeG still performs poorly in terms of both criteria, which matches the theoretical results in Table 1.

## Acknowledgements

The research of J. Cao, R. Jiang, and A. Mokhtari is supported in part by NSF Grant 2127697 and the NSF AI Institute for Foundations of Machine Learning (IFML) at UT Austin. The research of N. Abolfazli and E. Yazdandoost Hamedani is supported by NSF Grant 2127696.

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

# Supplementary material

## A  Additional Motivating Examples

The bilevel optimization problem in (1) provides a versatile framework that covers a broad class of optimization problems. In addition to the motivating examples provided in the main body of the paper, here we also provide a generic example of stochastic convex constrained optimization that can be formulated as (1). We further present a more general form of the examples covered in the main body.

*Stochastic convex optimization with many conic constraints:* Consider the following convex optimization problem

$$\min_{\mathbf{x} \in \mathbb{R}^d} \mathbb{E}[\tilde{f}(\mathbf{x}, \theta)] \qquad \text{s.t.} \quad h(\mathbf{x}, \xi) \in -\mathcal{K}, \ \forall \xi \in \Omega,$$

where $\mathcal{K} \subseteq \mathbb{R}^d$ is a closed convex cone. This problem can be formulated as a special case of (1) by letting $\tilde{g}(\mathbf{x}, \xi) = \frac{1}{2} d_{-\mathcal{K}}^2 (h(\mathbf{x}, \xi))$ where $d_{-\mathcal{K}}(\cdot) \triangleq \| \cdot - \mathcal{P}_{-\mathcal{K}}(\cdot)\|$ denotes the distance function and $\mathcal{P}_{-\mathcal{K}}(\cdot)$ denotes the projection map. Our proposed framework provides an efficient method for solving this class of problems when the projections onto $\mathcal{K}$ can be computed efficiently, while the projection onto the preimage $h^{-1}(-\mathcal{K}, \xi)$ is not practical, e.g., when $\mathcal{K}$ is the positive semidefinite cone, computing a projection onto the preimage set requires solving a nonlinear SDP.

### A.1  Lexicographic optimization

*Example 1 (over-parameterized regression)* can be generalized as a broader class of problem, which is known as lexicographic optimization [13] and uses the secondary loss to improve generalization. The problem can be formulated as the following stochastic simple bilevel optimization problem,

$$\min_{\boldsymbol{\beta} \in \mathbb{R}^d} \mathcal{L}(\boldsymbol{\beta}) \quad \text{s.t.} \quad \boldsymbol{\beta} \in \arg\min_{\theta \in \mathcal{Z}} \ell_{\text{tr}}(\theta) = \mathbb{E}_{\mathcal{D}_{tr}}[\ell(y, \hat{y}_\theta(\mathbf{x}))] \tag{17}$$

In general, the lower-level problem could have multiple optimal solutions and be very sensitive to small perturbations. To tackle the issue, we use a secondary criterion $\mathcal{L}(\cdot)$ to select some of the optimal solutions with our desired properties. For instance, we can find the optimal solutions with minimal $\ell_2$-norm by letting $\mathcal{L}(\boldsymbol{\beta}) = \|\boldsymbol{\beta}\|^2$, which is also known as *Lexicographic $\ell_2$ Regularization*.

### A.2  Lifelong learning

*Example 2 (dictionary learning)* is an instance of a popular framework known as lifelong learning, which can be formulated as follows,

$$\min_{\boldsymbol{\beta}} \frac{1}{n'} \sum_{i=1}^{n'} \ell\left(\langle \mathbf{x}_i', \boldsymbol{\beta} \rangle, y_i'\right) \qquad \text{s.t.} \qquad \sum_{(\mathbf{x}_i, y_i) \in \mathcal{M}} \ell(\langle \mathbf{x}_i, \boldsymbol{\beta} \rangle, y_i) \le \sum_{(\mathbf{x}_i, y_i) \in \mathcal{M}} \ell(\langle \mathbf{x}_i, \boldsymbol{\beta}^{(t-1)} \rangle, y_i) \tag{18}$$

In this problem, the objective is the training loss on the current tasks $\mathcal{D}_t = \{(\mathbf{x}_i', y_i')\}_{i=1}^{n'}$. While the constraint enforces that the model parameterized by $\boldsymbol{\beta}$ performs no worse than the previous one on the episodic memory $\mathcal{M}$ (i.e., data samples from all the past tasks).

In the paper, we discuss a variant of the problem above, where we slightly change the constraint and ensure that the current model also minimizes the error on the past tasks. It can be formulated as the following finite-sum/stochastic simple bilevel optimization problem [12],

$$\min_{\boldsymbol{\beta}} \frac{1}{n'} \sum_{i=1}^{n'} \ell\left(\langle \mathbf{x}_i', \boldsymbol{\beta} \rangle, y_i'\right) \qquad \text{s.t.} \quad \boldsymbol{\beta} \in \arg\min_{\mathbf{z}} \sum_{(\mathbf{x}_i, y_i) \in \mathcal{M}} \ell\left(\langle \mathbf{x}_i, \mathbf{z} \rangle, y_i\right). \tag{19}$$

## B  Supporting lemmas

### B.1  Proof of Lemma 4.1

Before we proceed to the proof for Lemma 4.1, we present the following technical lemma, which gives us an upper bound for a complex term appearing in the following analysis.

**Lemma B.1.** *Define $\rho_t = 1/(t+1)^\omega$ where $\omega \in (0,1]$ and $t \geq 1$. For all $t \geq 2$, let $\{s_t\}$ be a sequence of real numbers given by*

$$s_t = \sum_{\tau=2}^{t} \left( \rho_\tau \prod_{k=\tau}^{t} (1 - \rho_k) \right)^2.$$

*Then it holds that*

$$s_t \leq \frac{1}{(t+1)^\omega}. \tag{20}$$

*Proof.* We prove the result by induction. For $t = 2$, we can verify that

$$s_2 = \left( \frac{1}{3^\omega} \cdot \frac{3^\omega - 1}{3^\omega} \right)^2 \leq \frac{1}{3^{2\omega}} \leq \frac{1}{3^\omega}.$$

Now we suppose that the inequality in (20) holds when $t = T$ for some $T \geq 2$, i.e.,

$$s_T = \sum_{\tau=2}^{T} \left( \rho_\tau \prod_{k=\tau}^{T} (1 - \rho_k) \right)^2 \leq \frac{1}{(t+1)^\omega}.$$

First note that the sequence $\{s_t\}$ satisfies the following recurrence relation:

$$s_{T+1} = \sum_{\tau=2}^{T+1} \left( \rho_\tau \prod_{k=\tau}^{T+1} (1 - \rho_k) \right)^2 = (1 - \rho_{T+1})^2 \sum_{\tau=2}^{T+1} \left( \rho_\tau \prod_{k=\tau}^{T} (1 - \rho_k) \right)^2$$

$$= (1 - \rho_{T+1})^2 \left[ \sum_{\tau=2}^{T} \left( \rho_\tau \prod_{k=\tau}^{T} (1 - \rho_k) \right)^2 + \rho_{T+1}^2 \right]$$

$$= (1 - \rho_{T+1})^2 (s_T + \rho_{T+1}^2).$$

Moreover, since $\omega \in (0,1]$, we have $(T+2)^\omega - 1 \leq (t+1)^\omega$. Therefore, we obtain

$$s_{T+1} \leq \left( \frac{(T+2)^\omega - 1}{(T+2)^\omega} \right)^2 \left( \frac{1}{(t+1)^\omega} + \frac{1}{(T+2)^{2\omega}} \right)$$

$$\leq \frac{((T+2)^\omega - 1)(t+1)^\omega}{(T+2)^{2\omega}} \left( \frac{1}{(t+1)^\omega} + \frac{1}{(T+1)^{2\omega}} \right)$$

$$= \frac{(T+2)^\omega - 1}{(T+2)^{2\omega}} \frac{(T+1)^\omega + 1}{(T+1)^\omega}$$

$$= \frac{(T+2)^\omega (t+1)^\omega + (T+2)^\omega - 1 - (t+1)^\omega}{(T+2)^{2\omega}(t+1)^\omega}$$

$$\leq \frac{(T+2)^\omega (t+1)^\omega}{(T+2)^{2\omega}(t+1)^\omega} = \frac{1}{(T+2)^\omega}.$$

By induction, the inequality in (20) holds for all $t \geq 2$. $\qquad\square$

Now we proceed to prove Lemma 4.1.

*Proof of Lemma 4.1.* We show the proof of part (i) here. The proof of part (ii) is very similar to part (i). The first step is to reformulate $\mathbf{e}_t = \widehat{\nabla g}_t - \nabla g(\mathbf{x}_t)$ as the sum of a martingale difference sequence. For $t \geq 1$, by unrolling the reucurrence we have

$$\mathbf{e}_t = (1 - \beta_t)\mathbf{e}_{t-1} + \beta_t(\nabla \tilde{g}(\mathbf{x}_t, \xi_t) - \nabla g(\mathbf{x}_t))$$

$$\qquad + (1 - \beta_t)(\nabla \tilde{g}(\mathbf{x}_t, \xi_t) - \nabla \tilde{g}(\mathbf{x}_{t-1}, \xi_t) - (\nabla g(\mathbf{x}_t) - \nabla g(\mathbf{x}_{t-1})))$$

$$= \prod_{k=2}^{t} (1 - \beta_k)\mathbf{e}_1 + \sum_{\tau=2}^{t} \prod_{k=\tau}^{t} (1 - \beta_k)(\nabla \tilde{g}(\mathbf{x}_\tau, \xi_\tau) - \nabla \tilde{g}(\mathbf{x}_{\tau-1}, \xi_\tau) - (\nabla g(\mathbf{x}_\tau) - \nabla g(\mathbf{x}_{\tau-1}))$$

$$+ \sum_{\tau=2}^{t} \beta_\tau \prod_{k=\tau+1}^{t} (1 - \beta_k)(\nabla \tilde{g}(\mathbf{x}_\tau, \xi_\tau) - \nabla g(\mathbf{x}_\tau)).$$

$$\tag{21}$$

Thus, we can write $\mathbf{e}_t$ as the sum $\mathbf{e}_t = \sum_{\tau=1}^t \zeta_\tau$, where we define $\zeta_1 = \prod_{k=2}^t (1 - \beta_k)\mathbf{e}_1$ and

$$\zeta_\tau = \prod_{k=\tau}^t (1 - \beta_k)(\nabla\tilde{g}(\mathbf{x}_\tau, \xi_\tau) - \nabla\tilde{g}(\mathbf{x}_{\tau-1}, \xi_\tau) - (\nabla g(\mathbf{x}_\tau) - \nabla g(\mathbf{x}_{\tau-1}))) \tag{22}$$

$$+ \beta_\tau \prod_{k=\tau+1}^t (1 - \beta_k)(\nabla\tilde{g}(\mathbf{x}_\tau, \xi_\tau) - \nabla g(\mathbf{x}_\tau)) \tag{23}$$

for $\tau > 1$. Recall that $\mathbf{e}_1 = \nabla\tilde{g}(\mathbf{x}_t, \zeta_1) - \nabla g(\mathbf{x}_1)$. We observe that $\mathbb{E}[\zeta_\tau | \mathcal{F}_{\tau-1}] = 0$ where $\mathcal{F}_{\tau-1}$ is the $\sigma$-field generated by $\{\mathbf{x}_1, \xi_1, \ldots, \mathbf{x}_{\tau-1}, \xi_{\tau-1}\}$. Therefore, $\{\zeta_\tau\}_{\tau=1}^t$ is a martingale difference sequence.

Next, we derive upper bounds of $\|\zeta_\tau\|$. To begin with, we observe that for any $\tau = 1, 2, \ldots, t$,

$$\prod_{k=\tau}^t (1 - \beta_k) = \prod_{k=\tau}^t \left(1 - \frac{1}{(k+1)^\omega}\right) = \prod_{k=\tau}^t \frac{(k+1)^\omega - 1}{(k+1)^\omega} \leq \prod_{k=\tau}^t \frac{k^\alpha}{(k+1)^\omega} = \frac{\tau^\omega}{(t+1)^\omega}, \tag{24}$$

where we used the fact that $(k+1)^\omega - 1 \leq k^\omega$ in the last inequality. By using the above inequality, we can bound $\|\zeta_1\|$ as follows:

$$\|\zeta_1\| = \prod_{k=2}^t (1-\beta_k)\|\mathbf{e}_1\| \leq \frac{2^\omega}{(t+1)^\omega}\|\nabla\tilde{g}(\mathbf{x}_1, \xi_1) - \nabla g(\mathbf{x}_1)\| = \frac{2^\omega \sigma_1}{(t+1)^\omega}\frac{\|\nabla\tilde{g}(\mathbf{x}_1, \xi_1) - \nabla g(\mathbf{x}_1)\|}{\sigma_1}.$$

Define $c_1 = \frac{2^\omega \sigma_g}{(T+1)^\omega}$, then by Assumption 2.3(ii) we have $\mathbb{E}[\exp(\|\zeta_1\|^2/c_1^2)] \leq \exp(1)$. Moreover, for $\tau > 1$, by triangle inequality, $\|\zeta_\tau\|$ can be bounded by

$$\|\zeta_\tau\| \leq \prod_{k=\tau}^t (1 - \beta_k)(\|\nabla\tilde{g}(\mathbf{x}_\tau, \xi_\tau) - \nabla\tilde{g}(\mathbf{x}_{\tau-1}, \xi_\tau)\| + \|(\nabla g(\mathbf{x}_\tau) - \nabla g(\mathbf{x}_{\tau-1})\|)$$

$$+ \beta_\tau \prod_{k=\tau+1}^t (1 - \beta_k)\|\nabla\tilde{g}(\mathbf{x}_\tau, \xi_\tau) - \nabla g(\mathbf{x}_\tau)\|$$

$$\leq 2L_g\|\mathbf{x}_\tau - \mathbf{x}_{\tau-1}\|\prod_{k=\tau}^t (1 - \beta_k) + \|\nabla\tilde{g}(\mathbf{x}_\tau, \xi_\tau) - \nabla g(\mathbf{x}_\tau)\|\beta_\tau \prod_{k=\tau+1}^t (1 - \beta_k)$$

$$= 2L_g\gamma_\tau D \prod_{k=\tau}^t (1 - \beta_k) + \|\nabla\tilde{g}(\mathbf{x}_\tau, \xi_\tau) - \nabla g(\mathbf{x}_\tau)\|\beta_\tau \prod_{k=\tau+1}^t (1 - \beta_k) \tag{25}$$

$$\leq 2L_g D\beta_\tau \prod_{k=\tau}^t (1 - \beta_k) + \frac{3^\omega}{3^\omega - 1}\|\nabla\tilde{g}(\mathbf{x}_\tau, \xi_\tau) - \nabla g(\mathbf{x}_\tau)\|\beta_\tau \prod_{k=\tau}^t (1 - \beta_k)$$

$$= \left(2L_g D + \frac{3^\omega}{3^\omega - 1}\|\nabla\tilde{g}(\mathbf{x}_\tau, \xi_\tau) - \nabla g(\mathbf{x}_\tau)\|\right)\beta_\tau \prod_{k=\tau}^t (1 - \beta_k)$$

$$= \left(2L_g D + \frac{3^\omega \sigma_g}{3^\omega - 1}\frac{\|\nabla\tilde{g}(\mathbf{x}_\tau, \xi_\tau) - \nabla g(\mathbf{x}_\tau)\|}{\sigma_g}\right)\beta_\tau \prod_{k=\tau}^t (1 - \beta_k)$$

Define $c_\tau = (2L_g D + \frac{3^\omega \sigma_g}{3^\omega - 1})\beta_\tau \prod_{k=\tau}^t (1 - \beta_k)$. Note that if we have $\mathbb{E}[\exp(X_1^2/c_1^2)] \leq 1$ and $\mathbb{E}[\exp(X_2^2/c_2^2)] \leq 1$, then we have $\mathbb{E}[\exp((X_1 + X_2)^2/(c_1 + c_2)^2)] \leq 1$ [43]. Thus, we have $\mathbb{E}[\exp(\|\zeta_\tau\|^2/c_\tau^2)] \leq \exp(1)$ for all $1 \leq \tau \leq t$. Hence by proposition E.2, with probability $1 - \delta'$

$$\|\mathbf{e}_t\| \leq c \cdot \sqrt{\sum_{\tau=1}^t c_\tau^2 \log \frac{2d}{\delta'}} \tag{26}$$

where $c$ is an absolute constant, $d$ is the number of dimension, and $\sum_{\tau=1}^{T} c_\tau^2$ can be bounded by Lemma B.1 as follows,

$$
\begin{aligned}
\sum_{\tau=1}^{t} c_\tau^2 = c_1^2 + \sum_{\tau=2}^{t} c_\tau^2 &= \frac{2^{2\omega}\sigma_g^2}{(T+1)^{2\omega}} + (2L_g D + \frac{3^\omega}{3^\omega - 1}\sigma_g)^2 \sum_{\tau=2}^{T}(\beta_\tau \prod_{k=\tau}^{T}(1-\beta_k))^2 \\
&\leq \frac{2^{2\omega}\sigma_g^2}{(T+1)^{2\omega}} + \frac{(2L_g D + \frac{3^\omega}{3^\omega-1}\sigma_g)^2}{(t+1)^\omega} \\
&\leq \frac{((\sqrt{2})^\omega \sigma_g)^2}{(t+1)^\omega} + \frac{(2L_g D + \frac{3^\omega}{3^\omega-1}\sigma_g)^2}{(t+1)^\omega} \\
&\leq \frac{2(2L_g D + \frac{3^\omega}{3^\omega-1}\sigma_g)^2}{(t+1)^\omega}
\end{aligned}
\tag{27}
$$

where the last inequality follows from the fact that $(\sqrt{2})^\omega \leq 3^\omega/(3^\omega - 1)$ for any $\omega \in (0,1]$. Combining (26) and (27), we have with probability at least $1 - \delta'$,

$$
\|\nabla g(\mathbf{x}_t) - \widehat{\nabla g}_t\| \leq c\sqrt{2}(2L_g D + \frac{3^\omega}{3^\omega-1}\sigma_g)(t+1)^{-\omega/2}\sqrt{\log(2d/\delta')} \stackrel{\text{def}}{=} K_{1,t}
\tag{28}
$$

Similarly with probability at least $1 - \delta'$,

$$
|g(\mathbf{x}_t) - \hat{g}_t| \leq c\sqrt{2}(2L_l D + \frac{3^\omega}{3^\omega-1}\sigma_l)(t+1)^{-\omega/2}\sqrt{\log(2d/\delta')} \stackrel{\text{def}}{=} K_{0,t}
\tag{29}
$$

and with probability at least $1 - \delta'$,

$$
\|\nabla f(\mathbf{x}_t) - \widehat{\nabla f}_t\| \leq c\sqrt{2}(2L_f D + \frac{3^\omega}{3^\omega-1}\sigma_f)(t+1)^{-\omega/2}\sqrt{\log(2d/\delta')} \stackrel{\text{def}}{=} K_{2,t}
\tag{30}
$$

where $c$ is an absolute constant and $d$ is the dimension of vectors. We can use union bound to obtain that these three inequalities hold for at least probability $1 - 3\delta' = 1 - \delta$. For simplicity, we define constant $A_1^\omega$ and $A_0^\omega$ such that,

$$
A_1^\omega (t+1)^{-\omega/2}\sqrt{\log(6d/\delta)} = K_{1,t} \quad \text{and} \quad A_0^\omega (t+1)^{-\omega/2}\sqrt{\log(6d/\delta)} = K_{0,t}
\tag{31}
$$

and similarly $A_2^\omega (t+1)^{-\omega/2}\sqrt{\log(6d/\delta)} = K_{2,t}$. $\qquad\square$

## B.2 Proof of Lemma 4.2

*Proof.* Let us define $t_0 \triangleq \lfloor t/q \rfloor$ for any $t \in \{0, \ldots, T-1\}$, then whenever $t = t_0 q$ according to the Algorithm 2 a full batch of sample gradients are selected, hence, $\widehat{\nabla g}_t = \nabla g(\mathbf{x}_t)$; otherwise, the error of computing a sample gradient can be expressed as follows

$$
\epsilon_{t,i} = \frac{1}{S}(\nabla g_{\mathcal{S}(i)}(\mathbf{x}_t) - \nabla g_{\mathcal{S}(i)}(\mathbf{x}_{t-1}) - \nabla g(\mathbf{x}_t) + \nabla g(\mathbf{x}_{t-1})),
\tag{32}
$$

where $i$ is the index with $\mathcal{S}(i)$ denoting the $i$-th random component function selected at iteration $t$. Furthermore, from the update rule of $x_t$ we have $\|\mathbf{x}_t - \mathbf{x}_{t-1}\| = \gamma_t \|\mathbf{s}_{t-1} - \mathbf{x}_{t-1}\| \leq D\gamma$ for any $t \geq 0$, therefore,

$$
\begin{aligned}
\|\epsilon_{t,i}\| &\leq \frac{1}{S}(\|\nabla g_i(\mathbf{x}_t) - \nabla g_i(\mathbf{x}_{t-1})\| + \|\nabla g(\mathbf{x}_t) - \nabla g(\mathbf{x}_{t-1})\|) \\
&\leq \frac{2L_g}{S}\|\mathbf{x}_t - \mathbf{x}_{t-1}\| \leq \frac{2L_g D\gamma}{S},
\end{aligned}
\tag{33}
$$

for all $t \in \{t_0 + 1, \ldots, t_0 + q\}$ and $i \in \{1, \ldots, S\}$. On the other hand, from the update of $\widehat{\nabla g}_t$ and (32) we have that for any $t \neq t_0 q$, $\widehat{\nabla g}_t - \nabla g(\mathbf{x}_t) = \widehat{\nabla g}_{t-1} - \nabla g(\mathbf{x}_{t-1}) + \sum_{i=1}^{S} \epsilon_{t,i}$. Therefore, by continuing the recursive relation and taking the norm from both sides of the equality we obtain

$$
\begin{aligned}
\|\widehat{\nabla g}_t - \nabla g(\mathbf{x}_t)\| &= \|\widehat{\nabla g}(\mathbf{x}_{t_0}) - \nabla g(\mathbf{x}_{t_0}) + \sum_{j=t_0+1}^{t}\sum_{i=1}^{S}\epsilon_{j,i}\| \\
&= \|\sum_{j=t_0+1}^{t}\sum_{i=1}^{S}\epsilon_{j,i}\|,
\end{aligned}
\tag{34}
$$

where the last equality follows from $\widehat{\nabla} g(\mathbf{x}_{t_0}) = \nabla g(\mathbf{x}_{t_0})$. Then by Proposition E.1, we have

$$\mathbb{P}(\|\widehat{\nabla} g_t - \nabla g(\mathbf{x}_t)\| \geq \lambda) \leq 4\exp(-\frac{\lambda^2}{4S(t-t_0)\frac{4L_g^2 D^2\gamma^2}{S^2}}) \leq 4\exp(-\frac{\lambda^2}{16L_g^2 D^2\gamma^2}), \qquad (35)$$

where the last inequality follows from the fact $S = \sqrt{n}$ and $t - t_0 \leq q = \sqrt{n}$. By setting $\lambda = (4L_g D\gamma\sqrt{\log(4/\delta')})$ for some $\delta' \in (0, 1)$, we have with probability at least $1 - \delta'$,

$$\|\widehat{\nabla} g_t - \nabla g(\mathbf{x}_t)\| \leq 4L_g D\gamma\sqrt{\log(4/\delta')}. \qquad (36)$$

Similarly, with probability at least $1 - \delta'$,

$$|\hat{g}_t - g(\mathbf{x}_t)| \leq 4L_l D\gamma\sqrt{\log(4/\delta')}, \qquad (37)$$

and with probability $1 - \delta'$,

$$\|\widehat{\nabla} f_t - \nabla f(\mathbf{x}_t)\| \leq 4L_f D\gamma\sqrt{\log(4/\delta')}. \qquad (38)$$

Then by union bound and $\delta = 3\delta'$, we show these three equalities hold with probability $1 - \delta$.

$\square$

## B.3 Proof of Lemma 4.3

*Proof.* Let $\mathbf{x}_g^*$ be any point in $\mathcal{X}_g^*$, i.e., any optimal solution of the lower-level problem. By definition, we have $g\left(\mathbf{x}_g^*\right) = g^*$. Since $g$ is convex and $g^* \leq g\left(\mathbf{x}_0\right)$, we have

$$g\left(\mathbf{x}_0\right) - g\left(\mathbf{x}_t\right) \geq g^* - g\left(\mathbf{x}_t\right) = g\left(\mathbf{x}_g^*\right) - g\left(\mathbf{x}_t\right) \geq \left\langle \nabla g\left(\mathbf{x}_t\right), \mathbf{x}_g^* - \mathbf{x}_t \right\rangle \qquad (39)$$

Add and subtract terms in inequality above, we have,

$$\left\langle \widehat{\nabla} g_t, \mathbf{x}_g^* - \mathbf{x}_t \right\rangle + \hat{g}_t - g(\mathbf{x}_0) \leq |\langle \widehat{\nabla} g_t - \nabla g(\mathbf{x}_t), \mathbf{x}_g^* - \mathbf{x}_t \rangle| + |\hat{g}_t - g(\mathbf{x}_t)| \qquad (40)$$

Considering the random hyperplane we used in (9), we want to prove the following inequality holds with high probability,

$$\left\langle \widehat{\nabla} g_t, \mathbf{x}_g^* - \mathbf{x}_t \right\rangle + \hat{g}_t - g(\mathbf{x}_0) \leq K_t \qquad (41)$$

Recall $K_t = K_{0,t} + DK_{1,t}$. And $K_{0,t}$ and $K_{1,t}$ were set as the high probability bounds of $\|\widehat{\nabla} g_t - \nabla g(\mathbf{x}_t)\|$ and $|\hat{g}_t - g(\mathbf{x}_t)|$ in Lemma 4.1 for Algorithm 1 or Lemma 4.2 for Algorithm 2. Then compare the two inequalities above and use Jensen's inequality, $|\langle \widehat{\nabla} g_t, \mathbf{x}_g^* - \mathbf{x}_t \rangle| + |\hat{g}_t - g(\mathbf{x}_0)| \leq K_t$ holds with high probability $1 - \delta$ for all $t \geq 0$. Hence, Lemma 4.3 holds with probability $1 - \delta$ for all $t \geq 0$. $\square$

## B.4 Improvement in one step

The following lemma characterizes the improvement of both the upper-level and lower-level objective values after one step of the algorithms.

**Lemma B.2.** *If Assumptions 2.1, 2.2, 2.3 are satisfied,*

*(i) For all $t \geq 0$, assume that $\mathcal{X}_g^* \subset \mathcal{X}_t$. Then we have*

$$\gamma_{t+1}\mathcal{G}(\mathbf{x}_t) \leq f(\mathbf{x}_t) - f(\mathbf{x}_{t+1}) + \gamma_{t+1}D\|\nabla f(\mathbf{x}_t) - \widehat{\nabla} f_t\| + \frac{L_f D^2\gamma_{t+1}^2}{2} \qquad (42)$$

*As a corollary, if $f$ is convex, we further have*

$$f(\mathbf{x}_{t+1}) - f^* \leq (1 - \gamma_{t+1})(f(\mathbf{x}_t) - f^*)) + \gamma_{t+1}D\|\nabla f(\mathbf{x}_t) - \widehat{\nabla} f_t\| + \frac{L_f D^2\gamma_{t+1}^2}{2}. \qquad (43)$$

*(ii) We have*

$$g(\mathbf{x}_{t+1}) - g(\mathbf{x}_0) \leq (1 - \gamma_{t+1})(g(\mathbf{x}_t) - g(\mathbf{x}_0)) + D\gamma_{t+1}(\|\nabla g(\mathbf{x}_t) - \widehat{\nabla g}_t\| + K_{1,t})$$
$$+ \gamma_{t+1}(\|g(\mathbf{x}_t) - \hat{g}_t\| + K_{0,t}) + \frac{L_g D^2 \gamma_{t+1}^2}{2}. \tag{44}$$

*Proof.* (i) Based on the $L_f$-smoothness of the expected function $f$ we show that $f(\mathbf{x}_{t+1})$ is bounded by

$$f(\mathbf{x}_{t+1}) \leq f(\mathbf{x}_t) + \nabla f(\mathbf{x}_t)^\top (\mathbf{x}_{t+1} - \mathbf{x}_t) + \frac{L_f}{2}\|\mathbf{x}_{t+1} - \mathbf{x}_t\|^2 \tag{45}$$

Replace the terms $\mathbf{x}_{t+1} - \mathbf{x}_t$ by $\gamma_{t+1}(\mathbf{s}_t - \mathbf{x}_t)$ and add and subtract the term $\gamma_{t+1}\widehat{\nabla f}_t^\top (\mathbf{s}_t - \mathbf{x}_t)$ to the right hand side to obtain,

$$f(\mathbf{x}_{t+1}) \leq f(\mathbf{x}_t) + \gamma_{t+1}(\nabla f(\mathbf{x}_t) - \widehat{\nabla f}_t)^\top (\mathbf{s}_t - \mathbf{x}_t) + \gamma_{t+1}\widehat{\nabla f}_t^\top (\mathbf{s}_t - \mathbf{x}_t) + \frac{L_f}{2}\|\mathbf{x}_{t+1} - \mathbf{x}_t\|^2 \tag{46}$$

By Lemma 4.3, $\mathcal{X}_g^* \subseteq \mathcal{X}_t$ with high probability $1 - \delta$, for all $t = 1, \ldots, T$. Note that if we define $\mathbf{s}_t' = \arg\max_{\mathbf{s} \in \mathcal{X}_t}\{\langle \nabla f(\mathbf{x}_t), \mathbf{x}_t - \mathbf{s}\rangle\}$. Recall that FW gap is $\mathcal{G}(\hat{\mathbf{x}}) = \max_{\mathbf{s} \in \mathcal{X}_g^*}\{\langle \nabla f(\hat{\mathbf{x}}), \hat{\mathbf{x}} - \mathbf{s}\rangle\}$. We can replace the inner product $\langle \widehat{\nabla f}_t, \mathbf{s}_t \rangle$ by its upper bound $\langle \widehat{\nabla f}_t, \mathbf{s}_t' \rangle$. Applying this substitution leads to

$$f(\mathbf{x}_{t+1}) \leq f(\mathbf{x}_t) + \gamma_{t+1}(\nabla f(\mathbf{x}_t) - \widehat{\nabla f}_t)^\top (\mathbf{s}_t - \mathbf{x}_t) + \gamma_{t+1}\widehat{\nabla f}_t^\top (\mathbf{s}_t' - \mathbf{x}_t) + \frac{L_f}{2}\|\mathbf{x}_{t+1} - \mathbf{x}_t\|^2$$

$$= f(\mathbf{x}_t) + \gamma_{t+1}(\nabla f(\mathbf{x}_t) - \widehat{\nabla f}_t)^\top (\mathbf{s}_t - \mathbf{x}_t) + \gamma_{t+1}(\widehat{\nabla f}_t - \nabla f(\mathbf{x}_t))^\top (\mathbf{s}_t' - \mathbf{x}_t)$$

$$- \gamma_{t+1}\nabla f(\mathbf{x}_t)^\top (\mathbf{x}_t - \mathbf{s}_t') + \frac{L_f}{2}\|\mathbf{x}_{t+1} - \mathbf{x}_t\|^2$$

$$\leq f(\mathbf{x}_t) + \gamma_{t+1}(\nabla f(\mathbf{x}_t) - \widehat{\nabla f}_t)^\top (\mathbf{s}_t - \mathbf{s}_t') - \gamma_{t+1}\mathcal{G}(\mathbf{x}_t) + \frac{L_f}{2}\|\mathbf{x}_{t+1} - \mathbf{x}_t\|^2$$

$$\leq f(\mathbf{x}_t) + \gamma_{t+1}D\|\nabla f(\mathbf{x}_t) - \widehat{\nabla f}_t\| - \gamma_{t+1}\mathcal{G}(\mathbf{x}_t) + \frac{L_f \gamma_{t+1}^2 D^2}{2}$$
$$\tag{47}$$

Rearrange the terms for the inequality above, we can obtain,

$$\gamma_{t+1}\mathcal{G}(\mathbf{x}_t) \leq f(\mathbf{x}_t) - f(\mathbf{x}_{t+1}) + \gamma_{t+1}D\|\nabla f(\mathbf{x}_t) - \widehat{\nabla f}_t)\| + \frac{L_f \gamma_{t+1}^2 D^2}{2} \tag{48}$$

As a simple corollary, since $\mathcal{G}(\mathbf{x}_t) \geq f(\mathbf{x}_t) - f^*$ when $f$ is convex, we have,

$$f(\mathbf{x}_{t+1}) - f^* \leq (1 - \gamma_{t+1})(f(\mathbf{x}_t) - f^*)) + \gamma_{t+1}D\|\nabla f(\mathbf{x}_t) - \widehat{\nabla f}_t\| + \frac{L_f D^2 \gamma_{t+1}^2}{2} \tag{49}$$

(ii) Based on the $L_g$-smoothness of the expected function $g$ we show that $g(\mathbf{x}_{t+1})$ is bounded by

$$g(\mathbf{x}_{t+1}) \leq g(\mathbf{x}_t) + \nabla g(\mathbf{x}_t)^\top (\mathbf{x}_{t+1} - \mathbf{x}_t) + \frac{L_g}{2}\|\mathbf{x}_{t+1} - \mathbf{x}_t\|^2 \tag{50}$$

Replace the terms $\mathbf{x}_{t+1} - \mathbf{x}_t$ by $\gamma_{t+1}(\mathbf{s}_t - \mathbf{x}_t)$ and add and subtract the term $\gamma_{t+1}\widehat{\nabla g}_t^\top (\mathbf{s}_t - \mathbf{x}_t)$ to the right-hand side to obtain,

$$g(\mathbf{x}_{t+1}) \leq g(\mathbf{x}_t) + \gamma_{t+1}(\nabla g(\mathbf{x}_t) - \widehat{\nabla g}_t)^\top (\mathbf{s}_t - \mathbf{x}_t) + \gamma_{t+1}\widehat{\nabla g}_t^\top (\mathbf{s}_t - \mathbf{x}_t) + \frac{L_g}{2}\|\mathbf{x}_{t+1} - \mathbf{x}_t\|^2 \tag{51}$$

Now by definition of the set $\mathcal{X}_t$, using $\langle \widehat{\nabla g}_t, \mathbf{s}_t - \mathbf{x}_t \rangle \leq g(\mathbf{x}_0) - \hat{g}_t + K_{0,t} + DK_{1,t}$. In addition, we could use Cauchy–Schwarz inequality to upper bound the second term. Then add and subtract $\gamma_{t+1}g(\mathbf{x}_0)$ on the right hand side to obtain,

$$g(\mathbf{x}_{t+1}) \leq g(\mathbf{x}_t) + \gamma_{t+1}(g(\mathbf{x}_0) - g(\mathbf{x}_t)) + \gamma_{t+1}D\|\nabla g(\mathbf{x}_t) - \widehat{\nabla g}_t\|$$
$$+ \gamma_{t+1}(g(\mathbf{x}_t) - \hat{g}_t) + \gamma_{t+1}(K_{0_t} + DK_{1_t}) + \frac{L_g}{2}\|\mathbf{x}_{t+1} - \mathbf{x}_t\|^2 \tag{52}$$

Then subtract $g(\mathbf{x}_0)$ on both sides,

$$g(\mathbf{x}_{t+1}) - g(\mathbf{x}_0) \leq (1 - \gamma_{t+1})(g(\mathbf{x}_t) - g(\mathbf{x}_0))$$
$$+ \gamma_{t+1}(D\|\nabla g(\mathbf{x}_t) - \widehat{\nabla g}_t\| + \|g(\mathbf{x}_t) - \hat{g}_t\| + K_{0,t} + DK_{1,t}) + \frac{L_g}{2}\|\mathbf{x}_{t+1} - \mathbf{x}_t\|^2 \tag{53}$$

and the claim in the lemma follows. $\qquad\square$

## C  Proof of Theorem for Algorithm 1

### C.1  Proof of Theorem 4.4

*Proof.* For **lower-level**, by Lemma B.2, we have

$$g(\mathbf{x}_{t+1}) - g(\mathbf{x}_0) \le (1 - \gamma_{t+1})(g(\mathbf{x}_t) - g(\mathbf{x}_0)) + D\gamma_{t+1}(\|\nabla g(\mathbf{x}_t) - \widehat{\nabla g}_t\| + K_{1,t})$$
$$+ \gamma_{t+1}(\|g(\mathbf{x}_t) - \hat{g}_t\| + K_{0,t}) + \frac{L_g D^2 \gamma_{t+1}^2}{2} \tag{54}$$

By Lemma 4.1, we have $\|\nabla g(\mathbf{x}_t) - \widehat{\nabla g}_t\| \le K_{1,t}$ and $\|g(\mathbf{x}_t) - \hat{g}_t\| \le K_{0,t}$ with probability $1 - \delta'$. Plug them in the inequality above to obtain,

$$g(\mathbf{x}_{t+1}) - g(\mathbf{x}_0) \le (1 - \gamma_{t+1})(g(\mathbf{x}_t) - g(\mathbf{x}_0)) + 2\gamma_{t+1}(DK_{1,t} + K_{0,t}) + \frac{L_g D^2 \gamma_{t+1}^2}{2}$$

$$\le (1 - \frac{1}{t+1})(g(\mathbf{x}_t) - g(\mathbf{x}_0)) \tag{55}$$

$$+ \frac{2(DA_1^1\sqrt{\log(6d/\delta')} + A_0^1\sqrt{\log(6/\delta')})}{(t+1)^{3/2}} + \frac{L_g D^2}{2(t+1)^2}$$

with probability $1 - \delta'$ for all $t$. Let $C_1 = 4(DA_1^1 + A_0^1)$ and $\delta = T\delta'$. Then, by applying the inequality recursively for $t = 1, \ldots, T-1$, we obtain that

$$g(\mathbf{x}_T) - g(\mathbf{x}_0) \le \left(1 - \frac{1}{T}\right)(g(\mathbf{x}_{T-1}) - g(\mathbf{x}_0)) + \frac{C_1/2\sqrt{\log(6d/\delta')}}{T^{3/2}} + \frac{L_g D^2}{2T^2}$$

$$= \prod_{t=1}^{T-1}\left(1 - \frac{1}{t+1}\right)(g(\mathbf{x}_0) - g(\mathbf{x}_0)) + \sum_{t=1}^{T-1} \frac{C_1/2\sqrt{\log(6d/\delta')}}{(t+1)^{3/2}} \prod_{i=t+1}^{T-1}(1 - \frac{1}{i+1})$$

$$+ \sum_{t=1}^{T-1} \frac{L_g D^2}{2(t+1)^2} \prod_{i=t+1}^{T-1}(1 - \frac{1}{i+1})$$

$$\le 0 + \frac{C_1/2\sqrt{\log(6d/\delta')}}{T} \sum_{t=1}^{T-1} \frac{1}{\sqrt{t+1}} + \frac{L_g D^2}{2T} \sum_{t=1}^{T-1} \frac{1}{t+1}$$

$$\le \frac{C_1\sqrt{\log(6d/\delta')}}{\sqrt{T}} + \frac{L_g D^2}{2T}(1 + \log T)$$

$$\le \frac{C_1\sqrt{\log(6td/\delta)}}{\sqrt{T}} + \frac{L_g D^2 \log T}{T}$$

$$\tag{56}$$

with probability $1 - \delta$.

For **upper-level**, by Lemma B.2, we have

$$f(\mathbf{x}_{t+1}) - f^* \le (1 - \gamma_{t+1})(f(\mathbf{x}_t) - f^*) + D\gamma_{t+1}(\|\nabla f(\mathbf{x}_t) - \widehat{\nabla f}_t\|) + \frac{L_f D^2 \gamma_{t+1}^2}{2} \tag{57}$$

By Lemma 4.1, we have $\|\nabla f(\mathbf{x}_t) - \widehat{\nabla f}_t\| \le \frac{A_2^1\sqrt{log(6d/\delta')}}{(t+1)^{1/2}}$ with probability $1 - \delta'$. Plug it in the inequality above to obtain,

$$f(\mathbf{x}_{t+1}) - f^* \le (1 - \frac{1}{t+1})(f(\mathbf{x}_t) - f^*) + \frac{DA_2^1\sqrt{\log(6d/\delta')}}{(t+1)^{3/2}} + \frac{L_f D^2}{2(t+1)^2} \tag{58}$$

with probability $1 - \delta^{'}$ for all $t$. Let $C_2 = 2DA_2^1$ and $\delta = T\delta'$. Then, by applying the inequality recursively for $t = 1, \ldots, T - 1$, we obtain that

$$
\begin{aligned}
f(\mathbf{x}_T) - f^* &\leq \left(1 - \frac{1}{T}\right)(f(\mathbf{x}_{T-1}) - f^*) + \frac{DA_2^1\sqrt{\log(6d/\delta')}}{T^{3/2}} + \frac{L_f D^2}{2T^2} \\
&= \prod_{t=1}^{T-1}\left(1 - \frac{1}{t+1}\right)(f(\mathbf{x}_0) - f^*) + \sum_{t=1}^{T-1}\frac{DA_2^1\sqrt{\log(6d/\delta')}}{(t+1)^{3/2}}\prod_{i=t+1}^{T-1}\left(1 - \frac{1}{i+1}\right) \\
&\quad + \sum_{t=1}^{T-1}\frac{L_f D^2}{2(t+1)^2}\prod_{i=t+1}^{T-1}\left(1 - \frac{1}{i+1}\right) \\
&\leq \frac{f(\mathbf{x}_0) - f^*}{T} + \frac{DA_2^1\sqrt{\log(6d/\delta')}}{T}\sum_{t=1}^{T-1}\frac{1}{\sqrt{t+1}} + \frac{L_f D^2}{2T}\sum_{t=1}^{T-1}\frac{1}{t+1} \\
&\leq \frac{f(\mathbf{x}_0) - f^*}{T} + \frac{2DA_2^1\sqrt{\log(6d/\delta')}}{\sqrt{T}} + \frac{L_f D^2}{2T}(1 + \log T) \\
&\leq \frac{f(\mathbf{x}_0) - f^*}{T} + \frac{2DA_2^1\sqrt{\log(6td/\delta)}}{\sqrt{T}} + \frac{L_f D^2 \log T}{T}
\end{aligned}
\tag{59}
$$

with probability $1 - \delta$. The theorem is obtained.

$\square$

## C.2  Proof of Theorem 4.5

*Proof.* For **lower-level**, by Lemma B.2, we have

$$
\begin{aligned}
g(\mathbf{x}_{t+1}) - g(\mathbf{x}_0) &\leq (1 - \gamma_{t+1})(g(\mathbf{x}_t) - g(\mathbf{x}_0)) + D\gamma_{t+1}(\|\nabla g(\mathbf{x}_t) - \widehat{\nabla}g_t\| + K_{1,t}) \\
&\quad + \gamma_{t+1}(\|g(\mathbf{x}_t) - \hat{g}_t\| + K_{0,t}) + \frac{L_g D^2 \gamma_{t+1}^2}{2}
\end{aligned}
\tag{60}
$$

By Lemma 4.1, we have $\|\nabla g(\mathbf{x}_t) - \widehat{\nabla}g_t\| \leq K_{1,t}$ and $\|g(\mathbf{x}_t) - \hat{g}_t\| \leq K_{0,t}$ with probability $1 - \delta^{'}$. Plug them in the inequality above to obtain,

$$
\begin{aligned}
g(\mathbf{x}_{t+1}) - g(\mathbf{x}_0) &\leq (1 - \gamma_{t+1})(g(\mathbf{x}_t) - g(\mathbf{x}_0)) + 2\gamma_{T+1}(DK_{1,t} + K_{0,t}) + \frac{L_g D^2 \gamma_{t+1}^2}{2} \\
&\leq (1 - \frac{1}{(T+1)^{2/3}})g(\mathbf{x}_t) - g(\mathbf{x}_0) \\
&\quad + \frac{2D(A_1^{2/3}\sqrt{\log(6d/\delta')} + A_0^{2/3}\sqrt{\log(6d/\delta')})}{(t+1)^{1/3}(T+1)^{2/3}} + \frac{L_g D^2}{2(T+1)^{4/3}}
\end{aligned}
\tag{61}
$$

with probability $1 - \delta^{'}$ for all $t$. Let $C_3 = 2(DA_1^{2/3} + A_0^{2/3})$. Then we can sum all the inequality up for all $t$ to obtain,

$$
\begin{aligned}
g(\mathbf{x}_{t+1}) - g(\mathbf{x}_0) &\leq (1 - \frac{1}{(T+1)^{2/3}})(g(\mathbf{x}_t) - g(\mathbf{x}_0)) + \frac{C_3\sqrt{\log(6d/\delta')}}{(t+1)^{1/3}(T+1)^{2/3}} + \frac{L_g D^2}{2(T+1)^{4/3}} \\
&\leq (1 - \frac{1}{(T+1)^{2/3}})(g(\mathbf{x}_t) - g(\mathbf{x}_0)) + \frac{C_3\sqrt{\log(6Td/\delta)} + L_g D^2/2}{(t+1)^{1/3}(T+1)^{2/3}}
\end{aligned}
\tag{62}
$$

By induction, we have for all $t \geq 1$,

$$
g(\mathbf{x}_{t+1}) - g(\mathbf{x}_0) \leq \frac{C_3\sqrt{\log(6Td/\delta)} + L_g D^2/2}{(T+1)^{1/3}}
\tag{63}
$$

with probability $1 - \delta$, where $\delta = T\delta^{'}$.

For **upper-level**, by Lemma B.2, we have

$$\gamma_{t+1}\mathcal{G}(\mathbf{x}_t) \leq f(\mathbf{x}_t) - f(\mathbf{x}_{t+1}) + \gamma_{t+1}D\|\nabla f(\mathbf{x}_t) - \widehat{\nabla f}_t\| + \frac{L_f\gamma_{t+1}^2 D^2}{2} \tag{64}$$

By Lemma 4.1, we have $\|\nabla f(\mathbf{x}_t) - \widehat{\nabla f}_t\| \leq \frac{A_2^{2/3}\sqrt{log(6d/\delta')}}{(t+1)^{1/3}}$ with probability $1 - \delta'$. Plug it and $\gamma_{t+1} = 1/(T+1)^{2/3}$ in inequality above to obtain,

$$\sum_{t=0}^{T-1}\gamma_{t+1}\mathcal{G}(\mathbf{x}_t) \leq f(\mathbf{x}_0) - f(\mathbf{x}_T) + D\sum_{t=0}^{T-1}\gamma_{t+1}\|\nabla f(\mathbf{x}_t) - \widehat{\nabla f}_t\| + \frac{L_f D^2}{2}\sum_{t=0}^{T-1}\gamma_{t+1}^2$$

$$\leq f(\mathbf{x}_0) - f(\mathbf{x}_T) + D\sum_{t=0}^{T-1}\frac{A_2^{2/3}\sqrt{log(6d/\delta')}}{(t+1)^{1/3}(T+1)^{2/3}} + \frac{L_f D^2}{2}\sum_{t=0}^{T-1}\frac{1}{(T+1)^{4/3}} \tag{65}$$

$$\leq f(\mathbf{x}_0) - f(\mathbf{x}_T) + \frac{3}{2}DA_2^{2/3}\sqrt{log(6d/\delta')} + \frac{L_f D^2}{2}\frac{1}{(T+1)^{1/3}}$$

Let $\mathbf{x}_{t^*} = \arg\min_{1 \leq t \leq T}\mathcal{G}(\mathbf{x}_t)$, then

$$\mathcal{G}(\mathbf{x}_{t^*}) \leq \frac{1}{\sum_{t=0}^{T-1}\gamma_{t+1}}\sum_{t=0}^{T-1}\gamma_{t+1}\mathcal{G}(\mathbf{x}_t)$$

$$\leq \frac{1}{(T+1)^{1/3}}(f(\mathbf{x}_0) - f(\mathbf{x}_T) + \frac{3}{2}DA_2^{2/3}\sqrt{log(6Td/\delta)} + \frac{L_f D^2}{2}\frac{1}{(T+1)^{1/3}}) \tag{66}$$

$$\leq \frac{1}{(T+1)^{1/3}}(f(\mathbf{x}_0) - \underline{f} + \frac{3}{2}DA_2^{2/3}\sqrt{log(6Td/\delta)} + \frac{L_f D^2}{2}\frac{1}{(T+1)^{1/3}})$$

with probability $1 - \delta$, where $\delta = T\delta'$. By letting $C_4 = \frac{3}{2}DA_2^{2/3}$, the theorem is obtained. □

# D  Proof of Theorem for Algorithm 2

## D.1  Proof of Theorem 4.6

*Proof.* For **lower-level** By Lemma B.2, we have

$$g(\mathbf{x}_{t+1}) - g(\mathbf{x}_0) \leq (1 - \gamma_{t+1})(g(\mathbf{x}_t) - g(\mathbf{x}_0)) + D\gamma_{t+1}(\|\nabla g(\mathbf{x}_t) - \widehat{\nabla g}_t\| + K_{1,t})$$
$$+ \gamma_{t+1}(\|g(\mathbf{x}_t) - \hat{g}_t\| + K_{0,t}) + \frac{L_g D^2\gamma_{t+1}^2}{2} \tag{67}$$

By Lemma 4.2, we have $\|\nabla g(\mathbf{x}_t) - \widehat{\nabla g}_t\| \leq 4L_g D\gamma\sqrt{\log(12/\delta')}$ and $\|g(\mathbf{x}_t) - \hat{g}_t\| \leq 4L_l D\gamma\sqrt{\log(12/\delta')}$ with probability $1 - \delta'$. Let $C_5 = 8D(DL_g + L_l)$ and $\delta = T\delta'$. Plug them in inequality above and let $\gamma_t = \gamma = \log T/T$ to obtain,

$$g(\mathbf{x}_{T+1}) - g(\mathbf{x}_0) \leq (1 - \gamma)(g(\mathbf{x}_T) - g(\mathbf{x}_0)) + (C_5\sqrt{\log(12/\delta')} + L_g D^2/2)\gamma^2 \tag{68}$$

with probability $1 - \delta/T$. Sum up the inequalities for all $1 \leq t \leq T$ to get,

$$g(\mathbf{x}_{T+1}) - g(\mathbf{x}_0) = (1 - \gamma)^T(g(\mathbf{x}_0) - g(\mathbf{x}_0)) + (C_5\sqrt{\log(12/\delta')} + L_g D^2/2)\gamma^2\sum_{k=1}^{T}(1 - \gamma)^k$$

$$\leq 0 + (C_5\sqrt{\log(12/\delta')} + L_g D^2/2)\gamma \leq \frac{(C_5\sqrt{\log(12T/\delta)} + L_g D^2/2)\log T}{T} \tag{69}$$

with probability $1 - \delta$.

For **upper-level**, by Lemma B.2, we have,

$$f(\mathbf{x}_T) - f^* \leq (1 - \gamma_T)f(\mathbf{x}_{T-1}) - f^* + D\gamma_T\|\nabla f(\mathbf{x}_{T-1}) - \widehat{\nabla f}_{t-1}\| + \frac{L_f D^2\gamma_T^2}{2} \tag{70}$$

Now we proceed by replacing the terms $\|\nabla f(\mathbf{x}_t) - \widehat{\nabla} f_t\|$ by its upper bounds from Lemma 4.2, i.e. $\|\nabla f(\mathbf{x}_t) - \widehat{\nabla} f_t\| \leq 4 L_f D \gamma \sqrt{\log{(12/\delta')}}$,

$$f(\mathbf{x}_T) - f^* \leq (1-\gamma)(f(\mathbf{x}_{T-1}) - f^*) + L_f D^2 \gamma^2 (4\sqrt{\log{(12/\delta')}} + 1/2) \tag{71}$$

with probability $(1-\delta')$. And we can choose $\delta = 3T\delta'$ Then by telescope, with $\gamma = \frac{\log T}{T}$, we can obtain,

$$f(\mathbf{x}_T) - f^* \leq (1-\gamma)^T (f(\mathbf{x}_0) - f^*) + (4\sqrt{\log{(12/\delta')}} + 1/2) L_f D^2 \gamma^2 \sum_{i=1}^{T} (1-\gamma)^i$$

$$\leq (1-\gamma)^T (f(\mathbf{x}_0) - f^*) + (4\sqrt{\log{(12/\delta')}} + 1/2) L_f D^2 \gamma \tag{72}$$

$$\leq \exp{(-\gamma T)}(f(\mathbf{x}_0) - f^*) + (4\sqrt{\log{(12/\delta')}} + 1/2) L_f D^2 \gamma$$

$$\leq (f(\mathbf{x}_0) - f^*)/T + (4\sqrt{\log{(12T/\delta)}} + 1/2) L_f D^2 \log T / T$$

with probability $1 - \delta$. Note that without loss of generality, we can assume $f(\mathbf{x}_0) - f^* \geq 0$. If it is less than 0, we can bound it by 0. By letting $C_6 = 5 L_f D^2$, the theorem is obtained.

$\square$

## D.2 Proof of Theorem 4.7

*Proof.* For **lower-level**, by Lemma B.2, we have

$$g(\mathbf{x}_{t+1}) - g(\mathbf{x}_0) \leq (1-\gamma_{t+1})(g(\mathbf{x}_t) - g(\mathbf{x}_0)) + D\gamma_{t+1}(\|\nabla g(\mathbf{x}_t) - \widehat{\nabla} g_t\| + K_{1,t})$$
$$+ \gamma_{t+1}(\|g(\mathbf{x}_t) - \hat{g}_t\| + K_{0,t}) + \frac{L_g D^2 \gamma_{t+1}^2}{2} \tag{73}$$

By Lemma 4.2, we have $\|\nabla g(\mathbf{x}_t) - \widehat{\nabla} g_t\| \leq 4 L_g D \gamma \sqrt{\log{(12/\delta')}}$ and $\|g(\mathbf{x}_t) - \hat{g}_t\| \leq 4 L_l D \gamma \sqrt{\log{(12/\delta')}}$ with probability $1 - \delta'$. Let $C_7 = 8D(DL_g + L_l)$ and $\delta = T\delta'$. Plug them in inequality above and let $\gamma_t = 1/\sqrt{T}$ to obtain,

$$g(\mathbf{x}_{t+1}) - g(\mathbf{x}_0) \leq (1 - \frac{1}{T^{1/2}})(g(\mathbf{x}_t) - g(\mathbf{x}_0)) + \frac{C_7\sqrt{\log{(12/\delta')}}}{T} + \frac{L_g D^2}{2T}$$

$$\leq (1 - \frac{1}{T^{1/2}})(g(\mathbf{x}_t) - g(\mathbf{x}_0)) + \frac{C_7\sqrt{\log{(12/\delta')}} + L_g D^2/2}{T} \tag{74}$$

with probability $1 - \delta/T$. Sum up the inequalities for all $t \geq 1$ to get,

$$g(\mathbf{x}_{t+1}) - g(\mathbf{x}_0) = (1 - \frac{1}{T^{1/2}})^t \mathbb{E}[g(\mathbf{x}_0) - g(\mathbf{x}_0)] + \frac{(C_7\sqrt{\log{(12/\delta')}} + L_g D^2/2)}{T} \sum_{k=1}^{t} (1 - \frac{1}{T^{1/2}})^k$$

$$\leq \frac{C_7\sqrt{\log{(12T/\delta)}} + L_g D^2/2}{T^{1/2}} \tag{75}$$

with probability $1 - \delta$.

For **upper-level**, by Lemma B.2, we have

$$\gamma_{t+1} \mathcal{G}(\mathbf{x}_t) \leq f(\mathbf{x}_t) - f(\mathbf{x}_{t+1}) + \gamma_{t+1} D \|\nabla f(\mathbf{x}_t) - \widehat{\nabla} f_t\| + \frac{L_f \gamma_{t+1}^2 D^2}{2} \tag{76}$$

By Lemma 4.2, we have $\|\nabla f(\mathbf{x}_t) - \widehat{\nabla} f_t\| \leq 4 L_f D \gamma \sqrt{\log{(12/\delta')}}$ with probability $1 - \delta'$. Plug it and $\gamma_{t+1} = 1/\sqrt{T}$ in inequality above to obtain,

$$\frac{1}{\sqrt{T}} \sum_{t=0}^{T-1} \mathcal{G}(\mathbf{x}_t) \leq f(\mathbf{x}_0) - f(\mathbf{x}_T) + D \sum_{t=0}^{T-1} \gamma_{t+1} \|\nabla f(\mathbf{x}_t) - \widehat{\nabla} f_t\| + \frac{L_f D^2}{2} \sum_{t=0}^{T-1} \gamma_{t+1}^2$$

$$\leq f(\mathbf{x}_0) - f(\mathbf{x}_T) + L_f D^2 (4\sqrt{\log{(12\delta')}} + 1/2) \tag{77}$$

Divide both sides by $\sqrt{T}$, we can get, Let $\mathbf{x}_{t^*} \triangleq \arg\min_{1 \leq t \leq T} \mathcal{G}(\mathbf{x}_t)$, then

$$\mathcal{G}(\mathbf{x}_{t^*}) \leq \frac{1}{T} \sum_{t=0}^{T-1} \mathcal{G}(\mathbf{x}_t) \leq \frac{f(\mathbf{x}_0) - \underline{f} + L_f D^2 (4\sqrt{\log(12T/\delta)} + 1/2)}{T^{1/2}} \tag{78}$$

with probability $1 - \delta$. By letting $C_8 = 5 L_f D^2$, the theorem is obtained. $\qquad\square$

## E   Azuma-Hoeffding-type inequalities

In this section, we present two useful vector versions of Azuma-Hoeffding-type concentration inequalities with uniform bound assumption or sub-gaussian assumption. They are crucial in our high probability analysis.

**Proposition E.1.** *(Pinelis and other 1994 [44], Theorem 3.5) Let $\zeta_1, \ldots, \zeta_t \in \mathbb{R}^d$ be a vector-valued martingale difference sequence w.r.t. a filtration $\{\mathcal{F}_t\}$, i.e. for each $\tau \in 1, \ldots, t$, we have $\mathbb{E}[\zeta_\tau | \mathcal{F}_{\tau-1}] = 0$. Suppose that $\|\zeta_\tau\| \leq c_\tau$ almost surely. Then $\forall t \geq 1$,*

$$P(\|\sum_{\tau=1}^{T} \zeta_\tau\| \geq \lambda) \leq 4 \exp(-\frac{\lambda^2}{4 \sum_{\tau=1}^{T} c_\tau^2}) \tag{79}$$

**Proposition E.2.** *(Jin et al. [45], Corollary 7) Let $\zeta_1, \ldots, \zeta_t \in \mathbb{R}^d$ be a vector-valued martingale difference sequence w.r.t. a filtration $\{\mathcal{F}_t\}$, i.e. for each $\tau \in 1, \ldots, t$, we have $\mathbb{E}[\zeta_\tau | \mathcal{F}_{\tau-1}] = 0$. Suppose that $\mathbb{E}[\exp(\|\zeta_\tau\|^2/c_\tau^2)] \leq \exp(1)$. Then there exists a absolute constant $c$ such that, for any $\delta > 0$, with probability at least $1 - \delta$,*

$$\|\sum_{\tau=1}^{T} \zeta_\tau\| \leq c \cdot \sqrt{\sum_{\tau=1}^{T} c_\tau^2 \log \frac{2d}{\delta}} \tag{80}$$

This proposition was also used in previous literature including [46] and [37]. It is common to use such martingale inequality to obtain some high-probability results recently.

## F   Experiment details

In this section, we include more details about the numerical experiments in Section 5. For completeness, we briefly introduce the update rules of aR-IP-SeG in [16] and DBGD in [13]. In the following, we use the notation $\Pi_{\mathcal{Z}}(\cdot)$ to denote the Euclidean projection onto the set $\mathcal{Z}$.
The aR-IP-SeG algorithm is given by,

$$
\begin{aligned}
\mathbf{y}_{t+1} &= \Pi_{\mathcal{Z}}(\mathbf{x}_t - \gamma_t(\nabla\tilde{f}(\mathbf{x}_t, \theta_t)) + \rho_t \nabla\tilde{g}(\mathbf{x}_t, \xi_t)) \\
\mathbf{x}_{t+1} &= \Pi_{\mathcal{Z}}(\mathbf{x}_t - \gamma_t(\nabla\tilde{f}(\mathbf{y}_t, \theta'_t)) + \rho_t \nabla\tilde{g}(\mathbf{y}_t, \xi'_t)) \\
\Gamma_{t+1} &= \Gamma_t + (\gamma_t \rho_t)^r \\
\bar{\mathbf{y}}_{t+1} &= \frac{\Gamma_t \bar{\mathbf{y}}_t + (\gamma_t \rho_t)^r \mathbf{y}_{t+1}}{\Gamma_{t+1}}
\end{aligned}
\tag{81}
$$

where $\gamma_t$ is the stepsize, $\rho_t$ is the regularization parameter, and $\bar{\mathbf{y}}_T$ is the output of the algorithm. In this experiment, we choose $\gamma_t = \gamma_0/(t+1)^{3/4}$ and $\rho_t = \rho_0(t+1)^{1/4}$ for some constants $\gamma_0$ and $\rho_0$. The DBGD-sto is a stochastic version of DBGD, which simply replaces the gradients in DBGD with stochastic gradients. Although the stochastic version of DBGD does not have a theoretical guarantee, it has been used to solve stochastic simple bilevel optimization problems in [13], which worked pretty well empirically. Hence, we use it as a baseline for solving stochastic simple bilevel problems and compare it with our proposed algorithms. The DBGD algorithm is given by

$$\mathbf{x}_{k+1} = \mathbf{x}_k - \gamma_k \left( \nabla f(\mathbf{x}_k) + \lambda_k \nabla g(\mathbf{x}_k) \right)$$

where $\gamma_k$ is the stepsize and we set $\lambda_k$ as

$$\lambda_k = \max\left\{\frac{\phi(\mathbf{x}_k) - \langle\nabla f(\mathbf{x}_k), \nabla g(\mathbf{x}_k)\rangle}{\|\nabla g(\mathbf{x}_k)\|^2}, 0\right\} \quad \text{and} \quad \phi(\mathbf{x}) = \min\left\{\alpha(g(\mathbf{x}) - \hat{g}), \beta\|\nabla g(\mathbf{x})\|^2\right\}$$

where $\alpha$ and $\beta$ are hyperparameters and $\hat{g}$ is a lower bound of $g^*$. In this experiment, we choose $\hat{g} = 0$. We also note that [13] only considered unconstrained simple bilevel optimization, i.e. $\mathcal{Z} = \mathbb{R}^d$. We further project $\mathbf{x}_t$ onto $\mathcal{Z}$ for each iteration to ensure the constraints are satisfied.

## F.1 Over-parameterized regression

**Dataset generation.** The original Wikipedia Math Essential dataset [30] composes of a data matrix of size $1068 \times 731$. We randomly select one of the columns as the outcome vector $\mathbf{b} \in \mathbb{R}^{1068}$ and the rest to be a new matrix $\mathbf{A} \in \mathbb{R}^{1068 \times 731}$. We set constraint parameter $\lambda = 10$ in this experiment.
**Initialization.** We run the algorithm, SPIDER-FW [40], with stepsize chosen as $\gamma_t = 0.1/(t+1)$ on the lower-level problem in (1). We terminate the process to get $\mathbf{x}_0$ as the initial point for both SBCGI 1 and SBCGF 2 after $10^5$ stochastic oracle queries.
**Implementation details.** We query stochastic oracle $9 \times 10^5$ times with stepsize $\gamma_t = 0.01/(t+1)$ and $\gamma = 10^{-5}$ for SBCGI 1 and SBCGF 2 with $K_t = 10^{-4}/\sqrt{t+1}$, respectively. In each iteration, we need to solve the following subproblem induced by the methods,

$$\min_{\mathbf{s}} \langle\nabla f(\boldsymbol{\beta}_k), \mathbf{s}\rangle \quad \text{s.t.} \quad \|\mathbf{s}\|_1 \leq \lambda, \langle\nabla g(\boldsymbol{\beta}_k), \mathbf{s} - \boldsymbol{\beta}_k\rangle \leq g(\boldsymbol{\beta}_0) - g(\boldsymbol{\beta}_k). \tag{82}$$

Introduce $\mathbf{s}^+, \mathbf{s}^- \geq 0$ such that $\mathbf{s} = \mathbf{s}^+ - \mathbf{s}^-$. Then we can reformulate the problem above as follows,

$$\begin{aligned}\min_{\mathbf{s}^+, \mathbf{s}^-} \quad & \langle\nabla f(\boldsymbol{\beta}_k), \mathbf{s}^+ - \mathbf{s}^-\rangle \\ \text{s.t. } & \mathbf{s}^+, \mathbf{s}^- \geq 0, \langle\mathbf{s}^+, \mathbf{1}\rangle + \langle\mathbf{s}^-, \mathbf{1}\rangle \leq \lambda, \langle\nabla g(\boldsymbol{\beta}_k), \mathbf{s}^+ - \mathbf{s}^- - \boldsymbol{\beta}_k\rangle \leq g(\boldsymbol{\beta}_0) - g(\boldsymbol{\beta}_k),\end{aligned} \tag{83}$$

where $\mathbf{1} \in \mathbb{R}^d$ is the all-one vector.
For aR-IP-SeG, we choose $\gamma_0 = 10^{-7}$ and $\rho_0 = 10^3$. For DBGD, we set $\alpha = \beta = 1$ and $\gamma_t = 10^{-6}$.

## F.2 Dictionary learning

**Dataset generation.** We generate 500 sparse coefficient vectors $\{\mathbf{x}_i\}_{i=1}^{250}$ and $\{\mathbf{x}_k'\}_{k=1}^{250}$ with 5 random nonzero entries, whose absolute values are drawn uniformly from $[0.2, 1]$. The entries of the random noise vectors $\{\mathbf{n}_i\}_{i=1}^{250}$ and $\{\mathbf{n}_k'\}_{k=1}^{250}$ are drawn from i.i.d. Gaussian distribution with mean 0 and standard deviation 0.01.
**Initialization.** We use a similar initialization procedure as [12], which consists of two phases. In the first phase, we run the standard Frank-Wolfe algorithm on both the variables $\mathbf{D} \in \mathbb{R}^{25 \times 40}$ and $\mathbf{X} \in \mathbb{R}^{40 \times 250}$ for $10^4$ iterations with the stepsize $\gamma_t = 1/\sqrt{t+1}$. Next, in the second phase, we fix the variable $\mathbf{X}$ and only update $\mathbf{D}$ using the Frank-Wolfe algorithm with exact line search for additional $10^4$ iterations to obtain $\hat{\mathbf{D}}$ and $\hat{\mathbf{X}}$ as the initial point for the full bilevel problem.
**Implementation Details.** We choose $\delta = 3$ in both problems (5). To be fair, all four algorithms start from the same initial point. We slightly modify the initial point by letting $\tilde{\mathbf{D}} \in \mathbb{R}^{25 \times 50}$ be the concatenation of $\hat{\mathbf{D}} \in \mathbb{R}^{25 \times 40}$ and 10 columns of all zeros vectors. Furthermore, we initialize another variable $\tilde{\mathbf{X}}$ randomly by choosing its entries from a standard Gaussian distribution and then normalizing each column to have a $\ell_1$-norm of $\delta$. We choose the stepsize as $\gamma_t = 0.1/(t+1)^{2/3}$ and $\gamma = 10^{-3}$ for our SBCGI 1 and SBCGF2 with $K_t = 0.01/(t+1)^{1/3}$, respectively. Empirically, we observe that taking one sample per iteration leads to a very unstable process in this problem. In this case, we choose a mini-batch of size 8 for SBCGI, aR-IP-SeG, and the stochastic version of DBGD. For each iteration, we will solve the following subproblem,

$$\min_{\tilde{\mathbf{D}}} \left\langle\nabla f_{\tilde{\mathbf{D}}}\left(\tilde{\mathbf{D}}_k, \tilde{\mathbf{X}}_k\right), \tilde{\mathbf{D}}\right\rangle \quad \text{s.t.} \quad \left\|\tilde{\mathbf{d}}_i\right\|_2 \leq 1, \left\langle\nabla g\left(\tilde{\mathbf{D}}_k\right), \tilde{\mathbf{D}} - \tilde{\mathbf{D}}_k\right\rangle \leq g\left(\tilde{\mathbf{D}}_0\right) - g\left(\tilde{\mathbf{D}}_k\right) \tag{84}$$

The above problem can be reformulated by using the KKT condition, which is equivalent to get a root of the following one-dimensional nonlinear equation involving $\lambda \geq 0$ :

$$\tilde{\mathbf{D}} = \Pi_{\mathcal{Z}}\left(\nabla f_{\tilde{\mathbf{D}}}\left(\tilde{\mathbf{D}}_k, \tilde{\mathbf{X}}_k\right) + \lambda\nabla g\left(\tilde{\mathbf{D}}_k\right)\right), \quad \left\langle\nabla g\left(\tilde{\mathbf{D}}_k\right), \tilde{\mathbf{D}} - \tilde{\mathbf{D}}_k\right\rangle = g\left(\tilde{\mathbf{D}}_0\right) - g\left(\tilde{\mathbf{D}}_k\right) \tag{85}$$

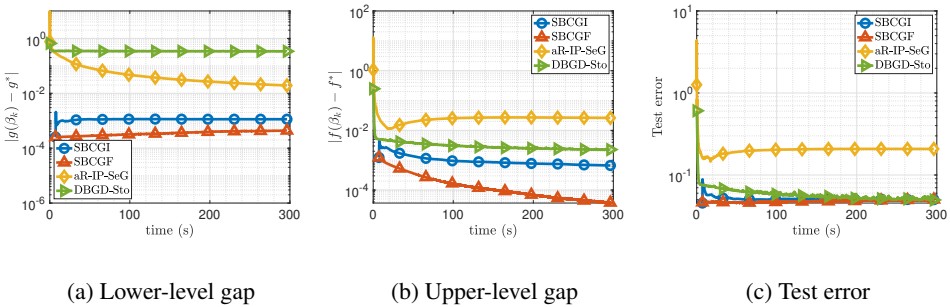

(a) Lower-level gap      (b) Upper-level gap      (c) Test error

Figure 3: Comparison of SBCGI, SBCGF, aR-IP-SeG, and DBGD-Sto for the over-parameterized regression problem

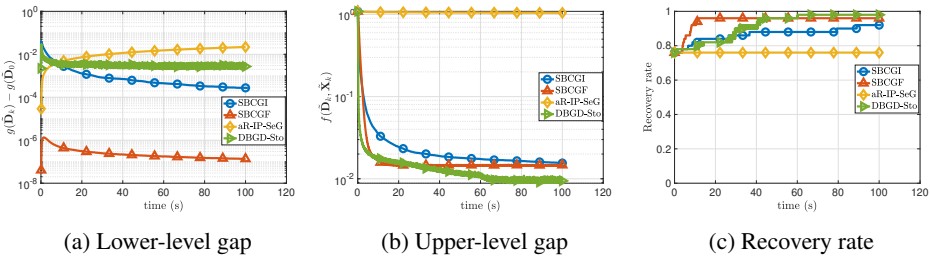

(a) Lower-level gap      (b) Upper-level gap      (c) Recovery rate

Figure 4: Comparison of SBCGI, SBCGF, aR-IP-SeG, and DBGD-Sto for solving the dictionary learning problem.

where the projection onto $\mathcal{Z} = \left\{ \tilde{\mathbf{D}} \in \mathbb{R}^{25 \times 50} : \left\| \tilde{\mathbf{d}}_i \right\|_2 \leq 1, i = 1, \ldots, 50 \right\}$ is equivalent to project each column on the Euclidean ball. In practice, the reformulated problem can be solved efficiently by MATLAB's root-finding solver.

For aR-IP-SeG, we choose $\gamma_0 = 10^{-4}$ and $\rho_0 = 1$. For the stochastic version of DBGD, we set $\alpha = \beta = 100$ and $\gamma_t = 5 \times 10^{-3}$.

Additional plots illustrating the comparison of the studied methods in terms of runtime rather than the number of sample used are provided in Figure 3 and Figure 4.

### F.3 Experiments with different random seeds

We further repeat the experiment 10 times with different random seeds to see more realizations of the stochastic algorithms. The results are reported in Figure 5 and Figure 6. The solid lines denote the average statistics over 10 trials of the algorithms. While the shaded regions surrounding each line reflect the span of all the random instances involved. Figure 5 and Figure 6 present similar results as Figure 1 and Figure 2, which eliminates the possibility of choosing a particularly good instance.

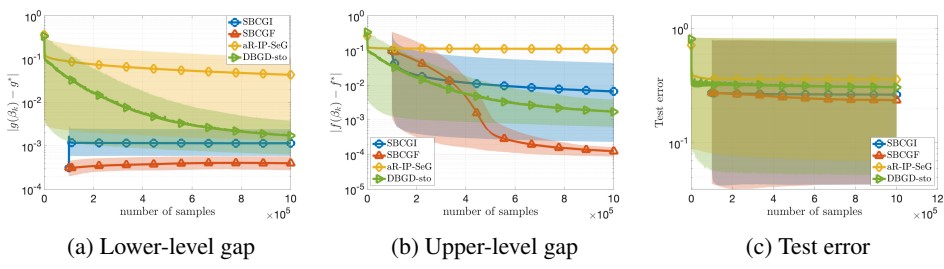

(a) Lower-level gap      (b) Upper-level gap      (c) Test error

Figure 5: Comparison of SBCGI, SBCGF, aR-IP-SeG, and DBGD-Sto for solving Problem (3) with 10 different random seeds

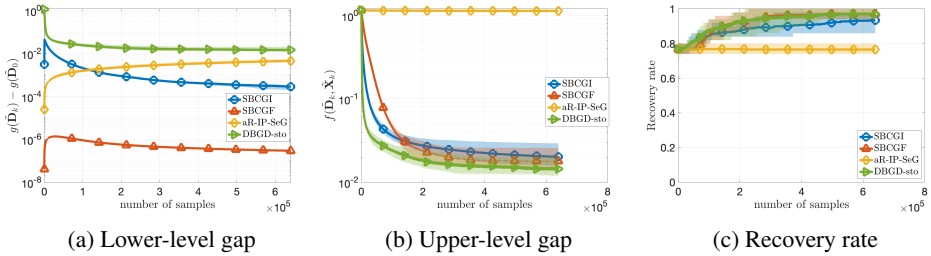

(a) Lower-level gap  (b) Upper-level gap  (c) Recovery rate

Figure 6: Comparison of SBCGI, SBCGF, aR-IP-SeG, and DBGD-Sto for solving Problem (5) with 10 different random seeds

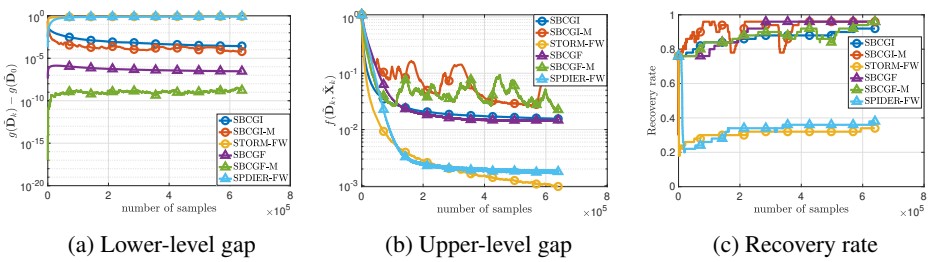

(a) Lower-level gap  (b) Upper-level gap  (c) Recovery rate

Figure 7: Comparison of SBCGI, SBCGF, SBCGI-M, SBCGF-M, STORM-FW, and SPIDER-FW for solving Problem (5).

### F.4 Importance of the right cutting plane

In this section, we numerically illustrate the importance of choosing the right cutting plane on *Example 2 (dictionary learning)*. Specifically, we compare our proposed methods with the ones without a cutting plane and with an unregularized cutting plane (without additional term $K_t$).

If we replace the stochastic cutting plane (9) with the unregularized cutting plane (8) in SBCGI 1 and SBCGF 2, then the algorithm usually fail at some point in the process, depending on the datasets and parameters chosen, based on our experimental observations. More specifically, algorithms' failure means that the subproblem of dictionary learning (85) is infeasible. So we slightly modify it by adding a checkpoint before solving the subproblem. If the subproblem is infeasible at the current iteration, then we choose the update direction $\mathbf{s}_t = \widehat{\nabla} g_t$. This adjustment prevents unnecessary interruptions during the process and enforce the algorithms to focus only on the lower-level problem when the subproblem is infeasible. We denote the modified algorithms SBCGI-M and SBCGF-M. Moreover, we also take SBCGI and SBCGF without cutting planes into consideration, denoted as STORM-FW and SPIDER-FW. In fact, in this case, the bilevel algorithms degenerate to single-level projection-free algorithms similar to algorithms in [37] and [40].

Figure 7 (a) indicates that SBCGI-M and SBCGF-M focus more on the lower-level problem due to the design of the algorithms and extremely unstable as we can see in Figure 7 (b)(c). While STORM-FW and SPIDER-FW only focus on the upper-level problem, which leads to terrible results on the lower-level gap and recovery rate.

