# OpenReview forum: "Projection-Free Methods for Stochastic Simple Bilevel Optimization with Convex Lower-level Problem"
_NeurIPS.cc/2023/Conference — NeurIPS 2023 poster_

### Official Review · Reviewer_oeu1 · 2023-06-15

**Soundness:** 4 excellent
**Presentation:** 4 excellent
**Contribution:** 3 good
**Rating:** 6
**Confidence:** 4

**Summary:**

A nice and clean result for stochastic simple bilevel optimization. The manuscript extends the algorithm design from the deterministic simple bilevel optimization with convex lower-level problem to the stochastic setting and demonstrate the sample complexity, which seems to be near-optimal (ignoring the log factor) when restricting the problem to classical stochastic optimization setting.

**Strengths:**

New results with good quality that are demonstrated in a nice and clean way. The motivation is very clear. It may push forward the study of general stochastic bilevel optimization when the lower-level is non-singleton.

**Weaknesses:**

The manuscript has no major flaws. The idea to add a $K_t$ and to estimate g using samples are rather straightforward though. The motivation for using variance reduction are not well-explained. Using variance reduction for lower level is to ensure that decaying $K_t$ makes sense. This step I doubt if it is possible to only use moving average estimator [Chen et al., 2021 NeurIPS, Closing the Gap: Tighter Analysis of Alternating Stochastic Gradient Methods for Bilevel Problems] on the lower-level problem to achieve the same effect. Using variance reduction for upper level is to obtain the current complexity bound. Otherwise, using vanilla Frank-wolfe on the upper level will result in larger complexity. But nevertheless, the manuscript already provides a full story. So that the algorithm would requires less hyperparameter to tune. See my questions below.



**Questions:**

1. Throughout your discussion of related literature and contributions, please accurately denote if the reference or contribution pertains to simple stochastic bilevel optimization or the broader general stochastic bilevel optimization. This applies to numerous instances, such as Line 139 and Line 45.

2. Regarding Line 103, is it $A\subset \mathbb{R}^m$ or $a_i\in\mathbb{R}^m$? There appears to be a typographical error.

3. How are $p$ and $q$ determined in continual dictionary learning? Do the old dataset $A$ and the new dataset $A^\prime$ have any correlation? If uncorrelated, why merge them? If correlated, would the results still hold?

4. In the 'Random set for the subproblem' subsection, is there a missing $\hat $ on $g(x_0)$? How is $g(x_0)$ known? This query also applies to equations (8)(9) and Algorithms 1 and 2.

5. Lemma 4.1 seems to miss a space in "c, (d is ". Additionally, why does the estimation error in Lemma 1 depend on the dimension? In other variance reduction papers, like STORM and SPIDER, the estimation error is generally dimension-independent.

6. In Theorem 1 and 2, $\mathcal{G}$ is not defined anywhere in the main context.

7. My understanding of general bilevel optimization implies variance reduction aims for $O(\epsilon^{-4})$ complexity bound in the nonconvex regime ($O(\epsilon^{-3})$ with variance reduction). However, Chen et al., 2021 [NeurIPS, closing the gap, tighter analysis] indicated that variance reduction isn't essential for achieving optimal complexity in strong and convex settings. Moreover, for nonconvex settings, an $O(\epsilon^{-4})$ rate can still be obtained without variance reduction. Therefore, I question whether variance reduction is genuinely necessary when $\hat{\nabla g}$ can be adaptively estimated using a moving average estimator, possibly achieving the same rate. This could be an avenue for future consideration.

**Limitations:**

The authors have adequately addressed the limitations and there is no societal impact involved as it is a pure theoretical work.

---

> ### Author Rebuttal · Authors · 2023-08-09
>
> We thank reviewer oeu1 for the comments and feedback!
>
> **Q1 Why do we use STORM and SPIDER rather than a simpler moving average estimator?**
>
> **A1**  This is an excellent point. Since our framework utilizes a projection-free approach for stochastic bilevel optimization, we have endeavored to incorporate the best variance reduction techniques studied in the literature when applying the projection-free method to stochastic single-level optimization problems.
>
> In fact, [Mokhtari et al. 2020] developed a stochastic Frank-Wolfe method using a moving average for stochastic optimization. However, their method's complexity was worse than the approaches achieved by unbiased variance reduction techniques like STORM [Zhang et al. 2020] or SPIDER [Yurtsever et al. 2019]. Given that, for stochastic single-level projection-free methods, the use of a biased moving average estimator led to suboptimal guarantees, we decided to employ unbiased variance reduction techniques such as STORM and SPIDER.
>
> That being said, we concur with the reviewer's observation that utilizing a simpler moving average estimator with fewer parameters to tune would be advantageous. Nevertheless, we are not aware of any analysis, even in the context of single-level stochastic settings, that matches the bounds established by [Zhang et al. 2020] and [Yurtsever et al. 2019] for projection-free methods.  Perhaps the fact that  [Chen et al. 2021] considered an unconstrained problem was crucial in allowing them to establish a strong complexity bound for the moving average estimator.  Exploring the feasibility of attaining similar results for constrained problems using a projection-free method remains an intriguing avenue of research that warrants further investigation.
>
> References:
>
> - Mokhtari et al., Stochastic Conditional Gradient Methods: From Convex Minimization to Submodular Maximization, 2020.
> - Zhang et al., One Sample Stochastic Frank-Wolfe, 2020
> - Yurtsever et al., Conditional Gradient Methods via Stochastic Path-Integrated Differential Estimator, 2019
>
> ---
>
> **Q2 How are $p$ and $q$ determined in continual dictionary learning? Do the old dataset $A$ and the new dataset $A'$ have any correlation? If uncorrelated, why merge them? If correlated, would the results still hold?**
>
> **A2** We used synthetic data for this problem, with $p$ and $q$ able to take any values, provided that $p<q$. The old dataset $A$ and the new dataset $A'$ are both generated from the same source, which in our example is a uniform distribution. The primary concept behind lifelong learning is that we only have access to a portion of the data at the outset, with new data gradually becoming available over time. The correlation between the datasets is not a critical factor.
>
> ---
>
> **Q3 How is $g(x_{0})$ known?**
>
> **A3** In the finite sum setting, we can exactly compute $g(x_{0})$, and the additional cost of $n$ function evaluations will be dominated by the overall complexity.
>
> In the stochastic setting, we could use a large batch of samples to estimate $g(x_{0})$ with high precision at the beginning of the process. This additional operation will not affect the overall sample complexity of the proposed method, as the additional cost is negligible compared to the overall sample complexity.
> Specifically, we need to take a batch size of $b = \tilde{\mathcal{O}}(\epsilon^{-2})$ to form the estimate $\hat{g}(x_{0})$. Using Hoeffding's inequality for subgaussian random variables, we have the following bound:
> $|\hat{g}(x_0) - g(x_{0})| \leq \sqrt{2}\sigma_{l}(T+1)^{-\omega/2}\sqrt{\log(2/\delta)}$, with a probability of at least $1-\delta$, where $T$ is the maximum number of iterations. Comparing this with Lemma 4.1.3, we can further derive:
> $
> |\hat{g}(x_0) - g(x_{0})| \leq \sqrt{2}\sigma_{l}(T+1)^{-\omega/2}\sqrt{\log(2/\delta)}
> \leq \sqrt{2}(2L_{l}D+\frac{3^{\omega}}{3^{\omega}-1}\sigma_{l})(t+1)^{-\omega/2}\sqrt{\log(6/\delta})
> $, with a probability of at least $1-\delta$ for all $0\leq t \leq T$. Consequently, the introduced error term would be absorbed in $K_{0, t}$ (Line 175) and will not affect any parts of the analysis. We will add this point to the revised paper. Thanks for your comment.
>
> ---
>
> **Q4 Why does the estimation error in Lemma 4.1 depend on the dimension? In other variance reduction papers, like STORM and SPIDER, the estimation error is generally dimension-independent.**
>
> **A4** The main reason our bounds depend on dimension is due to the fact that we provide high-probability bounds. This contrasts with most works that establish guarantees in expectation, which are dimension independent. More precisely, our results demonstrate convergence in high probability under the assumption of sub-gaussian noise (Assumptions 2.2.3 and 2.3.3 in our paper).
>
> It is worth noting that there are other works that provide guarantees with high probability and are dimension independent; however, they require stronger assumptions, such as the boundedness of random variables with probability 1, as seen in [Fang et al. (2018)] and [Xie et al. (2019)]. In Appendix E of our paper, Proposition E.1 is utilized to prove results under the assumption of bounded random variables, while Proposition E.2 is used to prove results under the sub-gaussian noise assumption.
>
> References:
>
> - Fang et al., SPIDER: Near-Optimal Non-Convex Optimization via Stochastic Path Integrated Differential Estimator, 2018.
> - Xie et al., Efficient Projection-Free Online Methods with Stochastic Recursive Gradient, 2019.
> ---
>
> **Q5 In Theorem 1 and 2, $\mathcal{G}$ is not defined anywhere in the main context.**
>
> **A5** Thanks for your comment. The definition of $\mathcal{G}(\hat{x})$ is provided in Definition 2.1 which states that
>   $  \mathcal{G}(\hat{x}) \triangleq \max _{s \in \mathcal{X}_g^*}\{\langle\nabla f(\hat{x}), \hat{x}-s\rangle\}$. We will add a pointer to this definition in the statements of both theorems.
>
> ---
>
> **Other typos.**
> Thank the reviewer for pointing out the typos. We will fix them.

---

> > ### Comment · Reviewer_oeu1 · 2023-08-10
> >
> > Thanks for the nice response. It clarifies my questions. Good luck!

---

### Official Review · Reviewer_Xgt8 · 2023-07-03

**Soundness:** 2 fair
**Presentation:** 2 fair
**Contribution:** 2 fair
**Rating:** 4
**Confidence:** 4

**Summary:**

The paper presents a novel approach for solving stochastic simple bilevel optimization problems. The authors develop new methods that locally approximate the solution set of the lower-level problem using a stochastic cutting plane and then employ a conditional gradient update with variance reduction techniques to manage error from stochastic gradients.

The specific contributions are:
- For the convex upper-level function case, the proposed method requires ${\tilde{\mathcal{O}}}(\max ( 1/\epsilon_f^2, 1/\epsilon_g^2 ) )$ stochastic oracle queries to find a solution that is $\epsilon_f$-optimal for the upper-level and $\epsilon_g$-optimal for the lower-level. This is a substantial improvement over the previous complexity of $\mathcal{O}(\max(1/\epsilon_f^4, 1/\epsilon_g^4))$.
- For the non-convex upper-level function case, the method requires at most ${\tilde{\mathcal{O}}}(\max(1/\epsilon_f^3, 1/\epsilon_g^3))$ stochastic oracle queries to find an $(\epsilon_f, \epsilon_g)$-stationary point.
- In the finite-sum setting, the method requires ${\tilde{\mathcal{O}}}(\sqrt{n}/\epsilon)$ and ${\tilde{\mathcal{O}}}(\sqrt{n}/\epsilon^2)$ stochastic oracle calls for the convex and non-convex settings, respectively, where $\epsilon= \min(\epsilon_f,\epsilon_g)$.


**Strengths:**

- Originality: In this paper, the idea of using cutting planes to approximate the optimal solution set for solving deterministic simple bilevel optimization is extended to the stochastic version. Due to the stochastic nature of the problem, this generalization is not trivial.
- Quality: From a theoretical standpoint, the quality of the research is notable, with the authors demonstrating a substantial improvement over previous complexity results.
- Clarity: The overall structure of the paper is clear.
- Significance: This research provide lower complexity guarantees for stochastic simple bilevel optimization.

**Weaknesses:**

- The authors make an assumption that the constraint set $\mathcal Z$ is compact and convex, which may not hold for many real applications. This strong assumption limits the method's applicability.

- In their experiments, the authors employ the number of samples as the measure on the abscissa axis without providing a clear rationale for this choice. Furthermore, a comparative analysis of the time efficiency across various methods is missing, which should ideally be included to give a more comprehensive evaluation of their approach.

- The paper contains numerous typographical errors which, although not affecting the technical content, detract from the overall readability and professionalism of the paper.

- In Example 1, the authors do not clearly delineate how randomness is incorporated and why it necessitates the use of stochastic simple bilevel optimization as a modeling approach for this particular problem.

**Questions:**


- Line 15 and elsewhere: The authors discard the logarithmic terms in the complexity results. I suggest the authors use $\tilde{\mathcal{O}}$ to hide the logarithmic terms.
- Line 46: The authors state that their approach is ``tight". Why?
- Example 1: The paper doesn't clarify where the element of randomness lies, nor does it explain why the problem specifically requires a stochastic simple bilevel optimization model.
- Line 156, the formula (8): How to compute the exact $g(x_0)$?
- Line 158: Why is there $\hat{g}(x_0)$ instead of $g(x_0)$?
- Algorithm 1, Line 7; Line 160, the formula (9);  Line 163 and elsewhere: The notations $\mathcal{X}_t$, $\hat{\mathcal{X}}_t$, $\mathcal{X}_t^{S}$ should be unified.
- Line 177: The citation (7) should be (9).
- Line 226, the formula (16): The term $\sqrt{\log(6/\delta)}$ should be changed to $\sqrt{\log(6d/\delta)}$.
- The authors state that $\omega=1$ and $\omega=2/3$ are the ``best'' choice for the convex and nonconvex settings, respectively. How to ensure that the choices are the best?
- Lemma 4.2 and Lemma 4.3: The authors could swap the places of these two lemmas and rewrite Lemma 4.2 to be more precise.
- Line 238, the last sentence: add "we" before "establish".
- In numerical experiments: Why do the authors use the number of samples be the abscissa axis? The paper doesn't include a comparative analysis of the time efficiencies across different algorithms, an omission that needs clarification.
- Line 316: The matrix $D^{'*}$ is not clearly defined.
- In Supplementary material: Many formulas are numbered but not cited, please consider removing these numbers. Sometimes the paper doesn't provide an explanation for the validity of two adjacent formulas. An appropriate clarification would be helpful for understanding.
- Line 479: The $S$ in the formula should be changed to $s_2$.
- Line 480 and elsewhere: The use of both "t" and "T" in the paper could lead to confusion. To avoid this, the authors could consider replacing one of them, such as changing "T" to "M".
- Line 482 and elsewhere: The "$\alpha$" should be replaced by "$\omega$".
- Line 526: the citation (47) is not correct.
- Line 563, the formula (55) and elsewhere: The term $g(x_t)-g(x_0)$ should be $(g(x_t)-g(x_0))$.
- Line 565, the formulas and elsewhere: The subscript $t+1$ in $g(x_{t+1})$ in the formula does not match the subscript $t$ in $g(x_t)$ in the theorem. This situation occurs many times in the proofs of theorems.
- Line 567, the formula (57) and elsewhere: The Lipchitz constant $L_g$ should be replaced by $L_f$.
- Line 677: The number $40$ should be replaced by $50$.

---

> ### Author Rebuttal · Authors · 2023-08-09
>
> We thank reviewer Xgt8 for the comments and feedback!
>
> **Q1 The authors assume that the constraint set $\mathcal{Z}$ is compact and convex, which may not hold for many real applications.**
>
> **A1** While this assumption may not apply to certain real-world scenarios, it remains a common assumption in many constrained simple bilevel optimization papers. These include works like [Jalilzadeh et al. (2023)] and [Jiang et al. (2023)], also in constrained general bilevel optimization papers, exemplified by [Liu et al. (2021)] and [Shen et al. (2023)]. Generally, assuming a compact constraint set is a prevalent practice when dealing with projection-free (Frank-Wolfe) methods, as demonstrated in works such as [Zhang et al. (2020)] and [Yurtsever et al. (2019)].  Since our work also employs a projection-free framework, it necessitates the assumption.
>
> References:
>
> - Jalilzadeh et al., Stochastic Approximation for Estimating the Price of Stability in Stochastic Nash Games, 2023
> - Jiang et al., A conditional gradient-based method for simple bilevel optimization with convex lower-level problem, 2023
> - Liu et al., Towards Gradient-based Bilevel Optimization with Non-convex Followers and Beyond, 2021
> - Shen et al., On Penalty-based Bilevel Gradient Descent Method, 2023
> - Zhang et al., One Sample Stochastic Frank-Wolfe, 2020
> - Yurtsever et al., Conditional Gradient Methods via Stochastic Path-Integrated Differential Estimator, 2019
> ---
> **Q2 Why do the authors use the number of samples as the abscissa axis? The paper doesn't include a comparative analysis of the time efficiencies across different algorithms.**
>
> **A2** A widely used metric for comparing iterative methods in stochastic optimization is the number of required sample queries to achieve a specific accuracy. However, as the reviewer suggested, it is also possible to compare these methods based on their runtime if sample complexity is not the primary point of comparison. To address the reviewer's concern, we have included new plots in the attached PDF illustrating the comparison of the studied methods in terms of runtime.
>
> ---
> **Q3 In Example 1, the paper doesn't clarify where the element of randomness lies.**
>
> **A3** Example 1 presents a bilevel optimization problem, where the lower-level problem pertains to training loss, while the upper-level problem corresponds to testing loss. In both instances, we are confronted with an Empirical Risk Minimization loss, which can be regarded as stochastic optimization problems. This involves a uniform distribution across individual elements of the loss associated with each sample point. Therefore, this scenario can be viewed as a specific instance of the stochastic simple bilevel problem described in equation (1). We will clarify it in the revised paper.
>
> ---
> **Q4 In Line 46, the authors state that their approach is "tight". Why?**
>
> **A4** It is essential to note that the sample complexity bounds of our proposed methods in both convex and nonconvex settings match the optimal sample complexity of "stochastic single-level" optimization problems. Consequently, these bounds are also tight for the more generalized setting of "stochastic bilevel" optimization problems considered in this paper. We will add the point to the revised paper.
>
> ---
> **Q5 How to compute the exact $g(x_0)$?**
>
> **A5** In the finite sum setting, we can exactly compute $g(x_{0})$. In the stochastic setting, we could use a batch of samples to estimate $g(x_{0})$ with high precision at the beginning. This additional operation will not affect the overall sample complexity of the proposed method, as the additional cost is negligible compared to the overall sample complexity. Specifically, we need to take a batch size of $b = \tilde{\mathcal{O}}(\epsilon^{-2})$ to form the estimate $\hat{g}(x_{0})$. By Hoeffding's inequality for subgaussian random variables, we have the following bound: $|\hat{g}(x_0) - g(x_{0})|\leq\sqrt{2}\sigma_{l}(T+1)^{-\omega/2}\sqrt{\log(2/\delta)}$, with a probability of at least $1-\delta$, where $T$ is the maximum number of iterations. Comparing this with Lemma 4.1.3, we can further derive: $|\hat{g}(x_0) - g(x_{0})|\leq \sqrt{2}\sigma_{l}(T+1)^{ \omega/2}\sqrt{\log(2/\delta)} \leq \sqrt{2}(2L_{l}D+\frac{3^{\omega}}{3^{\omega}-1}\sigma_{l})(t+1)^{-\omega/2}\sqrt{\log(6/\delta})$, with a probability of at least $1-\delta$ for all $0\leq t \leq T$. Consequently, the introduced error term would be absorbed in $K_{0, t}$ (Line 175) and will not affect any parts of the analysis. We will add the point to the revised paper.
>
> ---
> **Q6 Line 158: Why is there $\hat{g}(x_0)$ instead of $g(x_0)$?**
>
> **A6** This is a typo and it should be $g(x_0)$. We will fix it in the revised paper.
>
> ---
> **Q7 Line 226, the formula (16): The term $\sqrt{\log(6/\delta)}$ should be changed to $\sqrt{\log(6d/\delta)}$.**
>
> **A7** Since function values are real values, the dimension is $d=1$ and hence the bound $\sqrt{\log(6d/\delta)}$ can be simplified as $\sqrt{\log(6/\delta)}$.
>
> ---
> **Q8 The authors state that $\omega = 1$ and $\omega = 2/3$ are the ``best'' choices for the convex and nonconvex settings. How to ensure that the choices are the best?**
>
> **A8** What we meant to convey was that the best overall iteration and sample complexity, based on our analysis, for the convex setting is achieved when $\omega = 1$. Moreover, again based on our analysis, the best upper bound for the nonconvex setting is obtained for $\omega = 2/3$. We will clarify it in the revised paper.
>
> ---
> **Q9 Line 316: The matrix $D'^\*$ is not clearly defined.**
>
> **A9** As mentioned in Line 315-317, the matrix $\tilde{D}^* \in \mathbb{R}^{25 \times 50}$ contains all 50 true basis vectors and we generate matrices $D^* \in \mathbb{R}^{25 \times 40}$ and $D'^\* \in \mathbb{R}^{25 \times 20}$ by randomly selecting $40$ and $20$ columns of  $\tilde{D}^*$, respectively.
>
> ---
> **Other typos.** Thanks for pointing out these typos. We will make sure to fix them in our revision.

---

> > ### Comment · Reviewer_Xgt8 · 2023-08-20
> >
> > Thanks for the rebuttal and clarification.

---

### Official Review · Reviewer_iBKW · 2023-07-07

**Soundness:** 3 good
**Presentation:** 3 good
**Contribution:** 3 good
**Rating:** 6
**Confidence:** 4

**Summary:**

This paper introduces novel projection-free optimization methods for solving a class of *simple* stochastic bilevel optimization problems. The proposed methods are based on the conditional gradient algorithm and use variance reduction techniques to approximate the solution set of the lower-level problem. The paper provides theoretical guarantees for the convergence of the proposed methods and analyzes their sample complexity. The methods are shown to be efficient for both convex (global convergence) and non-convex upper-level (local convergence) functions.

**Strengths:**

The paper introduces a new framework for solving a class of stochastic bilevel optimization problems, which is an important and challenging problem in optimization. The proposed methods are shown to be efficient in experiments and have theoretical guarantees.

**Weaknesses:**

I am concerned that the paper makes no reference to the game theory literature on the topic (general-sum Stackelberg games).  Related work on the topic would make the work accessible to a broader audience.



**Questions:**

Questions: Can you describe how we can see this optimization problem from a game theory perspective?

---

> ### Author Rebuttal · Authors · 2023-08-09
>
> We thank reviewer iBKW for the comments and feedback!
>
> **Q1 Can you describe how we can see this optimization problem from a game theory perspective?**
>
> **A1** As the reviewer has correctly pointed out, a Stackelberg game (SCG) can be viewed as a general bilevel program in which a leader aims to maximize their own gain by anticipating and influencing the equilibrium state at which followers settle by engaging in a game [Liu et al., 2021; Li et al., 2022].
>
> However, in our paper, we are addressing a (stochastic) "simple" bilevel optimization problem that cannot be viewed as a two-player game. To be more precise, in a general bilevel optimization problem, the action of the leader player can influence the problem faced by the follower, since the lower-level objective function depends on a secondary variable. In contrast to this scenario, in simple bilevel optimization, the lower-level problem remains fixed and does not vary based on the actions of another player. As a result, it is not clear how one can view this optimization problem as a two-player game.
>
>
> We will add the above discussion to the revised paper to better highlight the connection between our setting and game theory literature.
>
> References:
>
> - Risheng Liu, Jiaxin Gao, Jin Zhang, Deyu Meng, Zhouchen Lin, Investigating Bi-Level Optimization for Learning and Vision from a Unified Perspective: A Survey and Beyond, 2021.
> - Jiayang Li, Jing Yu, Qianni Wang, Boyi Liu, Zhaoran Wang, Yu (Marco) Nie, Differentiable Bilevel Programming for Stackelberg Congestion Games, 2022

---

> > ### Comment · Reviewer_iBKW · 2023-08-21
> >
> > I thank the authors for their time and effort.
> >
> > I maintain my score.

---

### Official Review · Reviewer_FAuV · 2023-07-11

**Soundness:** 3 good
**Presentation:** 2 fair
**Contribution:** 3 good
**Rating:** 6
**Confidence:** 2

**Summary:**

This paper proposes methods for solving nonconvex/convex-convex simple bilevel optimization problems. The proposed methods are proven to have better non-asymptotic convergence guarantee than existing works. Numerical experiments are done on two applications to verify the effectiveness of the proposed methods.

**Strengths:**

The main contribution of this work is the improved convergence rates for solving nonconvex/convex-convex simple bilevel optimization problems. The proposed methods use a cutting plane to approximate the solution set of the lower-level problem, of which the solution may not be unique. In order to have a more accurate approximation, the work proposes to use variance reduced estimators (STORM for stochastic setting and SPIDER for finite-sum setting) for gradient and function value estimations.

**Weaknesses:**

1. In the related work section, line 69~71 says that existing works on solving general bilevel optimization only consider strongly convex lower-level problems. This statement is not true. There have several works done on studying general bilevel optimization without strong convexity condition of lower-level problems. Just to name a few: (Liu et al., 2021), (Sow et al., 2022), (Shen et al., 2023), etc. Since these works are solving general bilevel optimization, they are potentially applicable to the simple bilevel optimization. I believe it is reasonable to provide a more detailed discussion on how these existing works are/aren't applicable to the target formulation of this draft.
2. I notice that the algorithm SBCGI uses only one sample data per iteration, and the sample data batch size in algorithm SBCGF is fixed to $\sqrt{n}$. In general, fixed batch size is often not very practical, especially for large dataset. I wonder if the authors have any insights on why the batch size cannot be set arbitrarily and be a dependency parameter in the convergence rate.
3. Typo: in line 73, one->on.



References:

Risheng Liu, Yaohua Liu, Shangzhi Zeng, Jin Zhang, Towards Gradient-based Bilevel Optimization with Non-convex Followers and Beyond, 2021.

Daouda Sow, Kaiyi Ji, Ziwei Guan, Yingbin Liang, A Primal-Dual Approach to Bilevel Optimization with Multiple Inner Minima, 2022.

Han Shen, Quan Xiao, Tianyi Chen, On Penalty-based Bilevel Gradient Descent Method, 2023.

**Questions:**

Please see the weaknesses section.

**Limitations:**

The fixed batch size limits the proposed methods from applying on large-scale datasets.

---

> ### Author Rebuttal · Authors · 2023-08-09
>
> We thank reviewer FAuV for the comments and feedback!
>
> **Q1 There are several works on general bilevel optimization problems without lower-level strong convexity. The authors should provide more detailed discussions on how these existing works are/aren't applicable to the target formulation of this draft.**
>
> **A1** We thank the reviewer for pointing out these references. Indeed, there are several recent works on general bilevel optimization problems without lower-level strong convexity, and we will add them to the discussions in related work.
> Since simple bilevel problems can be viewed as a special case of general bilevel problems (where the lower-level objective is fixed), in principle these existing methods could be applied to our setting as well. However, there will be some technical issues, which we detail below.
>
> - Some of the works [Liu et al., 2021; Sow et al., 2022; Chen et al., 2023; Lu and Mei, 2023] only consider deterministic bilevel problems, where one has access to the exact gradient of the upper-level and lower-level objectives. In comparison, in this paper we focus on stochastic bilevel problems and one of the main challenges is to properly control the stochastic noises present in our oracle.
>
> - To the best of our knowledge, the only existing works considering stochastic bilevel problems without lower-level strong convexity are [Shen et al., 2023] and [Huang, 2023]. Note that the authors in both papers assume that the lower-level objective satisfies the PL inequality, while we assume that the lower-level objective is convex. Due to the different assumptions, their results are not directly comparable to ours.
>
> - Finally, since these works consider a more general and challenging class of problems, we argue that their theoretical results when applied to our setting are necessarily weaker.
>
>
> We will add the above discussion to the related work section of our revised paper. Thanks for your comment.
>
> References:
>
> - Risheng Liu, Yaohua Liu, Shangzhi Zeng, Jin Zhang, Towards Gradient-based Bilevel Optimization with Non-convex Followers and Beyond, 2021.
> - Daouda Sow, Kaiyi Ji, Ziwei Guan, Yingbin Liang, A Primal-Dual Approach to Bilevel Optimization with Multiple Inner Minima, 2022.
> - Lesi Chen, Jing Xu, and Jingzhao Zhang, On Bilevel Optimization without Lower-level Strong Convexity, 2023.
> - Zhaosong Lu and Sanyou Mei, First-Order Penalty Methods for Bilevel Optimization, 2023.
> - Han Shen, Quan Xiao, Tianyi Chen, On Penalty-based Bilevel Gradient Descent Method, 2023.
> - Feihu Huang, On Momentum-based Gradient Methods for Bilevel Optimization with Nonconvex Lower-level, 2023.
>
> ---
>
> **Q2 SBCGI uses a fixed batch size of 1 and SBCGF uses a fixed batch size of $\sqrt{n}$. Why the batch sizes cannot be set arbitrarily and be a parameter in the convergence rate?**
>
> **A2**
> Note that SBCGI can be implemented with a batch size as small as $b=1$. However, this does not imply that the stepsize "has to be" $b=1$. In other words, the main advantage of SBCGI, compared to SBCGF, is its capability to be implemented with any mini-batch size, even as small as $b=1$. Therefore, for SBCGI, the batch size can be set arbitrarily, whereas for SBCGF, it must be $\sqrt{n}$. We will emphasize this point in the revised paper
>
> ---
>
> **Typos.**
>
> Thank you for pointing out this typo. We will fix all typos mentioned by all reviewers in the revised paper.

---

> > ### Comment · Reviewer_FAuV · 2023-08-16
> >
> > Thanks the authors for the response. My concerns are addressed. I increase my score to 6.

---

### Author Rebuttal · Authors · 2023-08-09

In response to Reviewer Xgt8, we included some additional plots in the attached PDF file comparing the studied methods in terms of runtime.

---

### Comment · Area_Chair_u9Hz · 2023-08-11
**Discussion period**

Dear reviewers and authors,

Thank you very much for your work on this submission and its evaluation. Now that the authors have responded to the reviews, I *strongly encourage* the reviewers to acknowledge the review, to look at other reviews and rebuttals for this submission, and to adjust their scores if needed. Thanks to those that have already done so.

Authors have the possibility to reply if further questions are needed, until the 16th.

Thank you very much to all,

Area Chair

---

### Decision · Program_Chairs · 2023-09-21

**Decision:**

Accept (poster)

**Comment:**

There was a bit of disagreement between the reviewers on this submission, but overall, the interest of this submission was underlined by all, and the reviewer in favor of the rejection did not participate in the discussion.